# *In vitro* and *in vivo* characterization of a recombinant rhesus cytomegalovirus containing a complete genome

Husam Taher[1☉], Eisa Mahyari[1,2☉], Craig Kreklywich[1], Luke S. Uebelhoer[1¤a], Matthew R. McArdle[1¤b], Matilda J. Moström[3], Amruta Bhusari, Michael Nekorchuk[1,2], Xiaofei E[4], Travis Whitmer[1¤c], Elizabeth A. Scheef[3], Lesli M. Sprehe[3], Dawn L. Roberts[5], Colette M. Hughes[1], Kerianne A. Jackson[1], Andrea N. Selseth[1], Abigail B. Ventura[1], Hillary C. Cleveland-Rubeor[1], Yujuan Yue[6], Kimberli A. Schmidt[6], Jason Shao[7], Paul T. Edlefsen[7], Jeremy Smedley[2], Timothy F. Kowalik[4], Richard J. Stanton[5], Michael K. Axthelm[2], Jacob D. Estes[1,2], Scott G. Hansen[1], Amitinder Kaur[3], Peter A. Barry[6], Benjamin N. Bimber[1,2], Louis J. Picker[1], Daniel N. Streblow[1], Klaus Früh[1], Daniel Malouli[1]*

1 Vaccine and Gene Therapy Institute, Oregon Health and Science University, Beaverton, Oregon, United States of America, 2 Oregon National Primate Research Center, Oregon Health and Science University, Beaverton, Oregon, United States of America, 3 Tulane National Primate Research Center, Tulane University, Covington, Louisiana, United States of America, 4 Department of Microbiology and Physiological Systems, University of Massachusetts Medical School, Worcester, Massachusetts, United States of America, 5 Division of Infection and Immunity, Cardiff University School of Medicine, Cardiff, United Kingdom, 6 Center for Comparative Medicine and Department of Medical Pathology, University of California, Davis, California, United States of America, 7 Statistical Center for HIV/AIDS Research and Prevention, Vaccine and Infectious Disease Division, Fred Hutchinson Cancer Research Center, Seattle, Washington, United States of America

☉ These authors contributed equally to this work.
¤a Current address: Department of Pediatrics, Oregon Health & Science University, Portland, Oregon, United States of America
¤b Current address: Department of Biochemistry, University of Utah, Salt Lake City, Utah, United States of America
¤c Current address: Merck & Co., Inc., Kenilworth, New Jersey, United States of America
* malouild@ohsu.edu

**Data Availability Statement:** All relevant data are within the paper and its Supporting Information files except for the sequences of the viral strains

## Abstract

Cytomegaloviruses (CMVs) are highly adapted to their host species resulting in strict species specificity. Hence, *in vivo* examination of all aspects of CMV biology employs animal models using host-specific CMVs. Infection of rhesus macaques (RM) with rhesus CMV (RhCMV) has been established as a representative model for infection of humans with HCMV due to the close evolutionary relationships of both host and virus. However, the only available RhCMV clone that permits genetic modifications is based on the 68–1 strain which has been passaged in fibroblasts for decades resulting in multiple genomic changes due to tissue culture adaptations. As a result, 68–1 displays reduced viremia in RhCMV-naïve animals and limited shedding compared to non-clonal, low passage isolates. To overcome this limitation, we used sequence information from primary RhCMV isolates to construct a full-length (FL) RhCMV by repairing all mutations affecting open reading frames (ORFs) in the 68–1 bacterial artificial chromosome (BAC). Inoculation of adult, immunocompetent, RhCMV-naïve RM with the reconstituted virus resulted in significant viremia in the blood

that were isolated in this study as well as the sequences for the re-annotated 68-1, 68-1.2 and FL-RhCMV BACs which are available from Genbank under the accession numbers: BaCMV 31282 (MT157321), BaCMV 34826 (MT157322), CyCMV 31709 (MT157323), JaCMV 24655 (MT157324), RhCMV 34844 (MT157328), RhCMV KF03 (MT157329), RhCMV UCD52 (MT157330), RhCMV UCD59 (MT157331), RhCMV 68-1 BAC (MT157325), RhCMV 68-1.2 BAC (MT157326) and FL-RhCMV BAC (MT157327)."https://www.ncbi.nlm.nih.gov/genbank/.

**Funding:** National Institute of Allergy and Infectious Diseases (NIAID) https://www.niaid.nih.gov/: P01 AI129859U42 PAB, AK, KF R01 AI095113 to LJP P01 AI094417 to LJP R37 AI054292 to LJP R01 AI059457 to KF OD023038 to MKA the National Institutes of Health Office of the Director https://www.nih.gov/about-nih/what-we-do/nih-almanac/office-director-nih: U42OD010426 to OHSU P51OD011092 to OHSU P51OD011104 to Tulane University the Eunice Kennedy Shriver National Institute of Child Health & Human Development (NICHD) https://www.nichd.nih.gov/: 4DP2HD075699 to AK the Bill & Melinda Gates Foundation https://www.gatesfoundation.org/: OPP1033121 to LJP OPP1108533 to LJP OPP1152430 to LJP The funders had no role in study design, data collection and analysis, decision to publish, or preparation of the manuscript.

**Competing interests:** OHSU and Drs. S.G.H., L.J. P., K.F. and D.M. have a significant financial interest in VirBiotechnology, Inc., a company that may have a commercial interest in the results of this research and technology. The potential individual and institutional conflicts of interest have been reviewed and managed by OHSU.

similar to primary isolates of RhCMV and furthermore led to high viral genome copy numbers in many tissues at day 14 post infection. In contrast, viral dissemination was greatly reduced upon deletion of genes also lacking in 68–1. Transcriptome analysis of infected tissues further revealed that chemokine-like genes deleted in 68–1 are among the most highly expressed viral transcripts both *in vitro* and *in vivo* consistent with an important immunomodulatory function of the respective proteins. We conclude that FL-RhCMV displays *in vitro* and *in vivo* characteristics of a wildtype virus while being amenable to genetic modifications through BAC recombineering techniques.

## Author summary

Human cytomegalovirus (HCMV) infections are generally asymptomatic in healthy immunocompetent individuals, but HCMV can cause serious disease after congenital infection and in individuals with immunocompromised immune systems. Since HCMV is highly species specific and cannot productively infect immunocompetent laboratory animals, experimental infection of rhesus macaques (RM) with rhesus CMV (RhCMV) has been established as a closely related animal model for HCMV. By employing the unique ability of CMV to elicit robust and lasting cellular immunity, this model has also been instrumental in developing novel CMV-based vaccines against chronic and recurring infections with pathogens such as the human immunodeficiency virus (HIV) and *Mycobacterium tuberculosis (Mtb)*. However, most of this work was conducted with derivatives of the 68–1 strain of RhCMV which has acquired multiple genomic alterations in tissue culture. To model pathogenesis and immunology of clinical HCMV isolates we generated a full-length (FL) RhCMV clone representative of low passage isolates. Infection of RhCMV-naïve RM with FL-RhCMV demonstrated viremia and tissue dissemination that was comparable to that of non-clonal low passage isolates. We further demonstrate that FL-RhCMV is strongly attenuated upon deletion of gene regions absent in 68–1 thus demonstrating the usefulness of FL-RhCMV to study RhCMV pathogenesis.

## Introduction

Chronic human cytomegalovirus (HCMV) infections are generally asymptomatic in healthy, immunocompetent individuals and seroprevalence ranges from approximately 45% in developed countries to almost 100% of the population in the developing world [1]. However, the virus can cause significant disease after congenital infection and in individuals with immunocompromised immune systems [2]. No vaccines against HCMV exist, and treatment with antiviral drugs can limit acute infections but cannot eliminate the persistent virus [3]. Cytomegaloviruses are double stranded DNA viruses belonging to the herpesvirus subfamily *Betaherpesvirinae* and have so far been exclusively found in mammals, mainly rodents and primates [4]. CMVs contain the largest genomes of all herpesviruses and current annotations predict upwards of 170 open reading frames (ORFs) for most species. Ribozyme profiling data suggests that the actual number of translated viral mRNAs is likely significantly higher [5], however only a subset of these produce high levels of protein during infection of fibroblasts [6,7]. Co-evolution of these viruses with their host species over millions of years has led to a sequence relationship between CMV species that generally mirrors that of their hosts while also resulting in strict species specificity [8,9]. Hence, HCMV does not replicate and is not

pathogenic in immunocompetent animals, and animal models of HCMV thus generally rely on studying infection of a given host with their respective animal CMV. The most commonly used models are mice, rats, guinea pigs and rhesus macaques (RM). The close evolutionary relationship of RM to humans (as compared to rodents) is mirrored in the evolutionary relationship of the rhesus CMV (RhCMV) genome to HCMV as the overall genomic organization is similar and most viral gene families are found in both CMV species [10].

Infection of RM with RhCMV has thus become a highly useful animal model for HCMV including a model for congenital infection [11]. In addition, RhCMV has been used extensively to explore the possibility of harnessing the unique immune biology of CMV as a novel vaccine strategy, in particular the ability to elicit and maintain high frequencies of effector memory T cells [12]. This work revealed not only that RhCMV-based vectors are remarkably effective in protecting RM against challenge with simian immunodeficiency virus (SIV), *Mtb* and *Plasmodium knowlesi* [13–16], but also uncovered a unique and unexpected ability of RhCMV to be genetically programmed to elicit CD8$^+$ T cells that differ in their MHC restriction [17,18]. Importantly, highly attenuated RhCMV vaccine vectors that display greatly reduced viremia, dissemination and shedding maintain the adaptive immune program and the ability to protect against pathogen challenge [19,20].

However, the vast majority of these immunological and challenge studies relied on a molecular clone of RhCMV that was derived from strain 68–1 which differs significantly from circulating RhCMV strains. The RhCMV strain 68–1 was originally isolated in 1968 from the urine of a healthy RM [21] and had been extensively passaged on fibroblasts for more than 30 years before being cloned as a BAC [22]. During this time, 68–1 has acquired multiple tissue culture adaptations [10] including an inversion in a genomic region homologous to the HCMV ULb' region. This inversion simultaneously deleted the genes Rh157.5 and Rh157.4 (UL128 and UL130 in HCMV), two members of the viral pentameric receptor complex (PRC), as well as three of six genes encoding chemokine-like proteins homologous to the HCMV UL146 family [23]. Similar to PRC-deficient HCMV, the loss of a functional PRC resulted in restricted cell tropism of 68–1 RhCMV *in vitro* [24,25]. PRC-dependent infection of non-fibroblast cells, such as epithelial and endothelial cells, was increased upon the insertion of the Rh157.5 and Rh157.4 genes obtained from the unrelated RhCMV 180.92 strain [25]. Furthermore, strain 68–1 showed reduced viremia and shedding compared to the low passage isolates UCD52 and UCD59 upon primary infection of RhCMV-seronegative RM [26]. UCD52, UCD59 and 180.92 have also been used in congenital infection studies [11,27,28]. However, UCD52 and UCD59 [29,30] represent non-clonal isolates that have been passaged on rhesus epithelial cells instead of fibroblasts, a culture method that preserves the PRC but can also lead to tissue culture adaptations [31,32]. The 180.92 strain was shown to consist of a mixture of a tissue culture adapted and a wildtype variant with the latter rapidly emerging as the dominant variant *in vivo* [33]. To study the role of specific viral genes in RhCMV infection and pathogenesis there is a need for the construction of a BAC-cloned RhCMV representative of primary isolates. In addition, such a tissue culture non-adapted, but genetically modifiable RhCMV clone would also be a useful tool to model HCMV-based vaccine development for live-attenuated candidates derived from clinical isolates [34].

Here, we describe the construction of such a BAC-cloned RhCMV genome in which all presumed mutations in 68–1 that result in altered ORFs were repaired thus potentially reflecting a clone of the original 68–1 isolate prior to tissue culture passage. We demonstrate that the resulting viral sequence, termed FL-RhCMV, is representative of contemporary RhCMV isolates from multiple primate centers. FL-RhCMV demonstrates *in vitro* growth characteristics resembling those reported for primary isolates of HCMV, including the rapid adaptation to tissue culture through the accumulation of mutations in the gene homologous to HCMV RL13.

Furthermore, we show that FL-RhCMV displays wildtype-like viremia in RhCMV-seronegative RM. The availability of the first RhCMV BAC clone containing a complete genome sequence granting the derived virus all characteristics of a circulating isolate will enable the selected modulation of tissue tropism, pathogenesis and immune stimulation. This is exemplified by our demonstration that the deletion of the RhCMV homologs of HCMV UL128, UL130 and UL146 profoundly impacted viral dissemination and proliferation during acute infection. Thus, we report the generation of a RhCMV BAC that represents a primary isolate and that can serve as a modifiable progenitor for studies using RhCMV as model for HCMV infection or HCMV-based vaccine vectors.

## Results

### Construction of a full length (FL) RhCMV BAC and *in vitro* characterization

Genetic modifications of CMV genomes are most conveniently performed using bacterial artificial chromosome (BAC) recombineering [35,36]. Unfortunately, the only existing RhCMV BAC was based on the 68–1 RhCMV strain, an extensively fibroblasts passaged isolate which, compared to circulating and low passage isolates, has acquired a large inversion in the region homologous to one end of the HCMV "unique long" ($U_L$) sequence of the genome (commonly referred to as the ULb' region), flanked by deletions of multiple ORFs on either side of the inversion [23,37,38]. When sequencing the clonal BAC of 68–1, we additionally identified multiple viral ORFs (Rh13.1, Rh61/Rh60, Rh152/Rh151, and Rh197), that contained point mutations predicted to result in frameshifts or premature terminations of the annotated proteins [10]. The frame shift mutation located in Rh61/Rh60 (UL36 in HCMV) has been shown to render the encoded inhibitor of extrinsic apoptosis non-functional [39]. To generate a wildtype like RhCMV genome we thus decided to perform step by step repairs of all ORFs that are mutated in 68–1.

By reversing the frameshift in Rh61/Rh60 and by inserting the missing PRC members Rh157.5 and Rh157.4 (UL128 and UL130 in HCMV) from the unrelated RhCMV strain 180.92, BAC-cloned 68–1 had been previously partially repaired, resulting in RhCMV clone 68–1.2 which exhibits broader cell tropism (**Fig 1, repair 0**) [25]. However, the 68–1.2 RhCMV genome sequence still differed significantly from the sequence of low passage RhCMV isolates due to additional mutations that were likely acquired during the prolonged tissue culture of the original 68–1 isolate. The inverted segment in the $U_L$-homologous region of 68–1 RhCMV was recently re-examined by amplifying and sequencing DNA from the original urine sample used for virus isolation in 1968 [40]. This work revealed the sequence of genes deleted in the $U_L$ region upon later passage of 68–1 including the homologs of UL128 and UL130 which showed substantial sequence variation compared to the corresponding genes of 180.92 used in the repaired 68–1.2. This is likely due to significant genetic diversity across strains for these genes in RhCMV [41]. To create a BAC that most closely resembles a clone of the original 68–1 primary urine isolate we therefore synthesized the entire gene region containing the inverted and missing genes in the $U_L$ region in two overlapping fragments that were then inserted into 68–1.2 RhCMV by homologous recombination (**Fig 1, repair 1**). In the next step, we used *en passant* recombination to repair two frameshift mutations in Rh13.1 (discussed below) (**Fig 1, repair 2**) and a point mutation resulting in truncation of Rh152/151 (**Fig 1 repair 3**) a close homologue of a viral Fc-gamma receptor encoded by HCMV ORF UL119/118 [42]. Subsequently, we repaired a nonsynonymous point mutation in Rh164, a homolog of the HCMV UL141, a multifunctional proteins involved in NK cell and innate antiviral defense evasion [43] (**Fig 1 repair 4**). We furthermore restored Rh167 (O14), an old

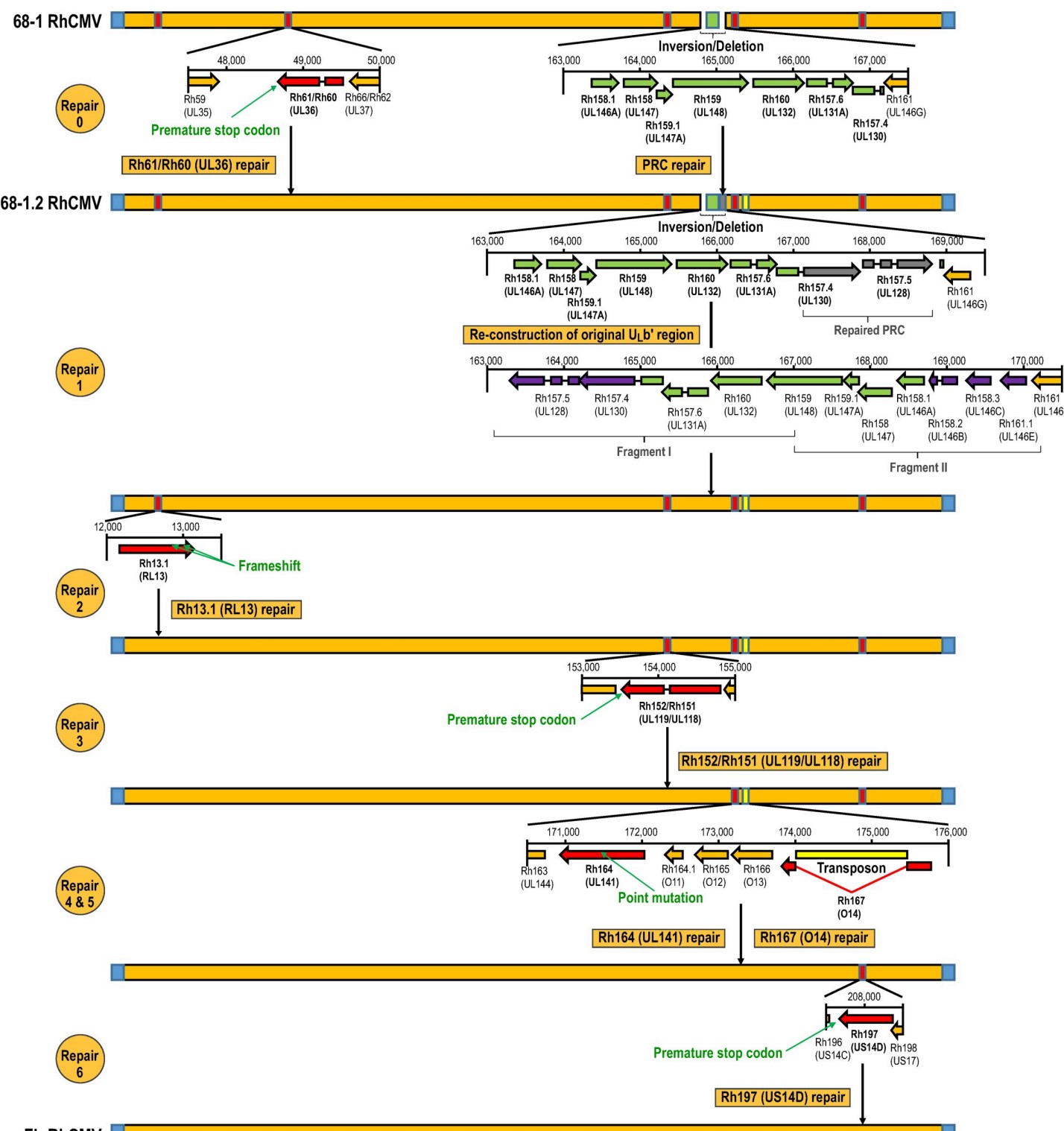

**Fig 1. Construction of FL-RhCMV.** The schematic depicts the repair steps performed to generate FL-RhCMV. Unaltered ORFs and the unmodified viral genome are shown in orange while the terminal repeats are indicated in blue. ORFs containing known mutations that were repaired in this study are highlighted in red. Genes contained in the acquired inversion in the ULb' region are shown in green, while genes lost in 68–1 but re-inserted into the genome during the repair are highlighted in purple. The transposon picked up during the generation of 68–1.2 RhCMV is highlighted in yellow and the 180.92 RhCMV PRC members used in the construction of 68–1.2 RhCMV are marked in grey. Repair 0: The frameshift resulting in a premature stop codon in Rh61/60 of 68–1 RhCMV was repaired and the two missing PRC members (Rh157.4 and Rh157.5) were inserted to generate 68–1.2 RhCMV as described previously [25]. Repair 1: Two DNA fragments combined spanning 6.9kb

corresponding to the genomic sequence of the ULb' homologous region in the circulating virus originally isolated from sample 68–1 [40] were synthesized. Three undefined bases in the published nucleotide sequence (KF011492) were taken from the consensus sequence of all sequenced low-passage RhCMV isolates. A synthetic DNA fragment spanning the region upstream of Rh157.5 (UL128) to Rh161 (UL146G) in its original orientation was used to replace the corresponding gene region in 68–1.2 RhCMV. The resulting construct maintains the repaired Rh61/60 gene while also containing the original isolate 68–1 genes Rh157.4 (UL128) and Rh157.5 (UL130) as well as the genes coding for the UL146 homologs Rh158.2, Rh158.3 and Rh161.1. Repair 2: Two previously described frameshift mutations in Rh13.1 [10] were repaired resulting in an intact Rh13.1 ORF. Repair 3: A premature stop codon in the viral Fcγ-Receptor homolog Rh152/151 [10] was repaired restoring the ORF to its original length. Repair 4: A nonsynonymous point mutations in Rh164 (UL141) initially predicted by us was confirmed by sequencing the original urine isolate. Hence, we restored the natural DNA sequence. Repair 5: Full genome sequencing of the 68–1.2 RhCMV BAC revealed that an *E. coli* derived transposon had inserted itself into the Rh167 ORF. The transposon was removed by *en passant* mutagenesis and the intact Rh167 ORF was restored. Repair 6: The US14 homolog Rh197 contained a premature stop codon which was repaired.

world NHP CMV specific protein of unknown function, which inadvertently acquired a transposon during the construction of the 68–1.2 RhCMV BAC (**Fig 1, repair 5**). Finally, we reverted a premature termination codon in the Rh197 ORF encoding for an HCMV US13/US14 homolog within the US12 gene family of seven-transmembrane spanning proteins (**S8 Fig**). While the specific function of the RhCMV gene is unknown, multiple HCMV US12 family members have been shown to be involved in NK cell evasion by selectively interfering with the plasma membrane expression of cellular proteins (**Fig 1, repair 6**) [44]. We confirmed the correct sequence of our BAC by restriction digest and next generation sequencing (NGS) and termed the final construct full length RhCMV (FL-RhCMV).

To characterize the phylogenetic relationship of the 68–1 derived FL-RhCMV BAC clone to related old world NHP CMV species and to ensure that FL-RhCMV is representative of circulating RhCMV isolates, we cultured and sequenced new isolates of RhCMV, cynomolgus CMV (CyCMV), Japanese macaque CMV (JaCMV) and baboon CMV (BaCMV) from two different US primate centers. We also performed NGS on viral DNA isolated from stocks of the extensively characterized RhCMV isolates UCD52 and UCD59 grown on epithelial cells and included these genome sequences into our analysis. For comparison, we included all NHP CMV sequences of complete or mostly complete genomes deposited into the GenBank database. As expected, FL-RhCMV clustered with all other RhCMV isolates and was more distantly related to the CMVs from other NHPs, with the evolutionary relationship of CMV species tracing the evolutionary relationship between their corresponding host species (**Fig 2**).

To ensure that FL-RhCMV contained the full ORFeome of all presently confirmed and predicted viral genes of circulating RhCMV strains, we compared the full annotation of FL-RhCMV with that of other old world NHP CMVs [37,45–47]. All RhCMV genomes lack an internal repeat sequence so that the genomic regions corresponding to the unique long ($U_L$) or the unique short ($U_S$) coding regions are fixed in a given orientation whereas HCMV and ChCMV genomes can freely switch between four isomeric forms (**S1 Fig**). Interestingly, the $U_S$-homologous region of BaCMV and Drill CMV (DrCMV) is fixed in the opposite orientation compared to RhCMV consistent with an isomer fixation event independent from the RhCMV lineage indicating that single isomers were fixed during the evolution of old world NHP CMVs on more than one occasion. A closer analysis of the genomes revealed that all primary RhCMV isolates without obvious deletions or inversions are predicted to contain the exact same ORFs in the same order. No strain-specific ORFs were identified based on our previously established RhCMV annotation [10]. FL-RhCMV contains all ORFs found in other RhCMVs indicating that the full genome content has been restored (**Fig 3**). A closer examination of full genome alignments of all known old world NHP CMV genome sequences additionally allowed us to further refine our previously established annotations with changes largely comprising reannotations of start codons and splice donor- and acceptor sites (**S1 Table**). Comparing the viral ORFeomes across old world NHP CMV species revealed a very high degree of conservation in the entire lineage of viruses so that the entire RhCMV

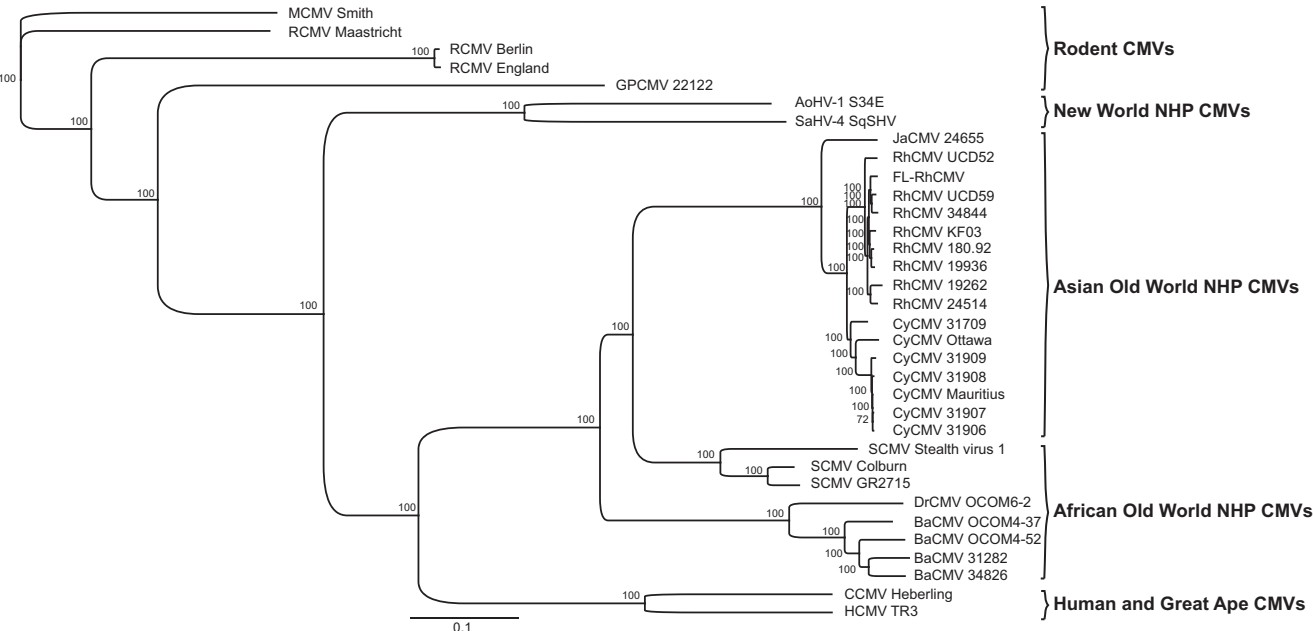

**Fig 2. Sequence relationship of FL-RhCMV with NHP CMVs.** A phylogenetic tree for FL-RhCMV and rodent and primate CMVs was constructed based on full genome alignments using the Geneious Prime Tree Builder application. Sequences previously published include RhCMV 180.92 [37] as well as the RhCMV isolates 19262, 19936 and 24514 and the Cynomolgus CMV isolates 31906, 31907, 31908 and 31909 [41]. We also included the published sequences for the CyCMV strains Ottawa [46] and Mauritius [47], the simian (African green monkey) CMV isolates Colburn [94], GR2715 [45] and stealth virus 1 [95] as well as the BaCMV strains OCOM4-37 [96] and OCOM4-52 [97] and the DrCMV strain OCOM6-2 [97]. For comparison we included the HCMV TR3 strain [34], the chimpanzee CMV strain Heberling [98] and the only two complete genome sequences of new world NHP CMVs, Aotine betaherpesvirus 1 (AoHV-1) strain S34E [99] and Saimiriine betaherpesvirus 4 (SaHV-4) strain SqSHV [100]. New genome sequences included in this alignment are as follows: the two RhCMV isolates 34844 and KF03, the CyCMV isolate 31709, the Japanese macaque CMV JaCMV 24655 and the two baboon CMVs BaCMV 31282 and 34826. These CMVs were isolated from fibroblast co-cultures of urine samples obtained from NHP housed either at the Oregon National Primate Research Center (ONPRC) or the Tulane National Primate Research Center (TNPRC). Also included in the alignment are the genomic sequences of the previously published RhCMV isolates UCD52 and UCD59 that originated at the UC Davis National Primate Research Center [29,30]. The rodent CMVs include the rat CMV (RCMV) isolates Maastricht [101], England [102] and Berlin [103], the guinea pig CMV (GPCMV) isolate 22122 [104] and the murine CMV (MCMV) strain Smith [105], which was used as an outgroup.

annotation can almost seamlessly be transferred to all related species. While our results are based on comparative genomics and hence need to be confirmed experimentally by mass spectrometry or ribozyme profiling, it is interesting to note that most ORFs that differ between NHP CMV species are due to gene duplication events that occurred in six different loci across the genome (**S2**–**S7 Figs**). Taken together we conclude that the FL-RhCMV clone represents a WT RhCMV genome. Since we relied, in part, on sequence information generated by re-analyzing the original urine sample used for virus isolation, we believe that this construct likely resembles a clone of the genomes contained in the original 68–1 isolate. While it is possible that FL-RhCMV still encodes additional unidentified mutations in non-coding regions, including introns for which tissue-culture adaptations have been described for HCMV [48], our phylogenic analysis clearly indicates that FL-RhCMV now contains the full complement of all ORFs identified in circulating isolates.

### *In vitro* characterization of FL-RhCMV

As we have reported earlier [10], one of the ORFs frequently mutated in passaged RhCMV and other old world NHP CMV isolates is the RL11 family member Rh13.1 (**Fig 3**). This ORF is homologous to HCMV RL13 which is often lost or mutated upon tissue culture passage of HCMV [32]. Loss of a functional RL13 protein was shown to result in more rapid cell to cell

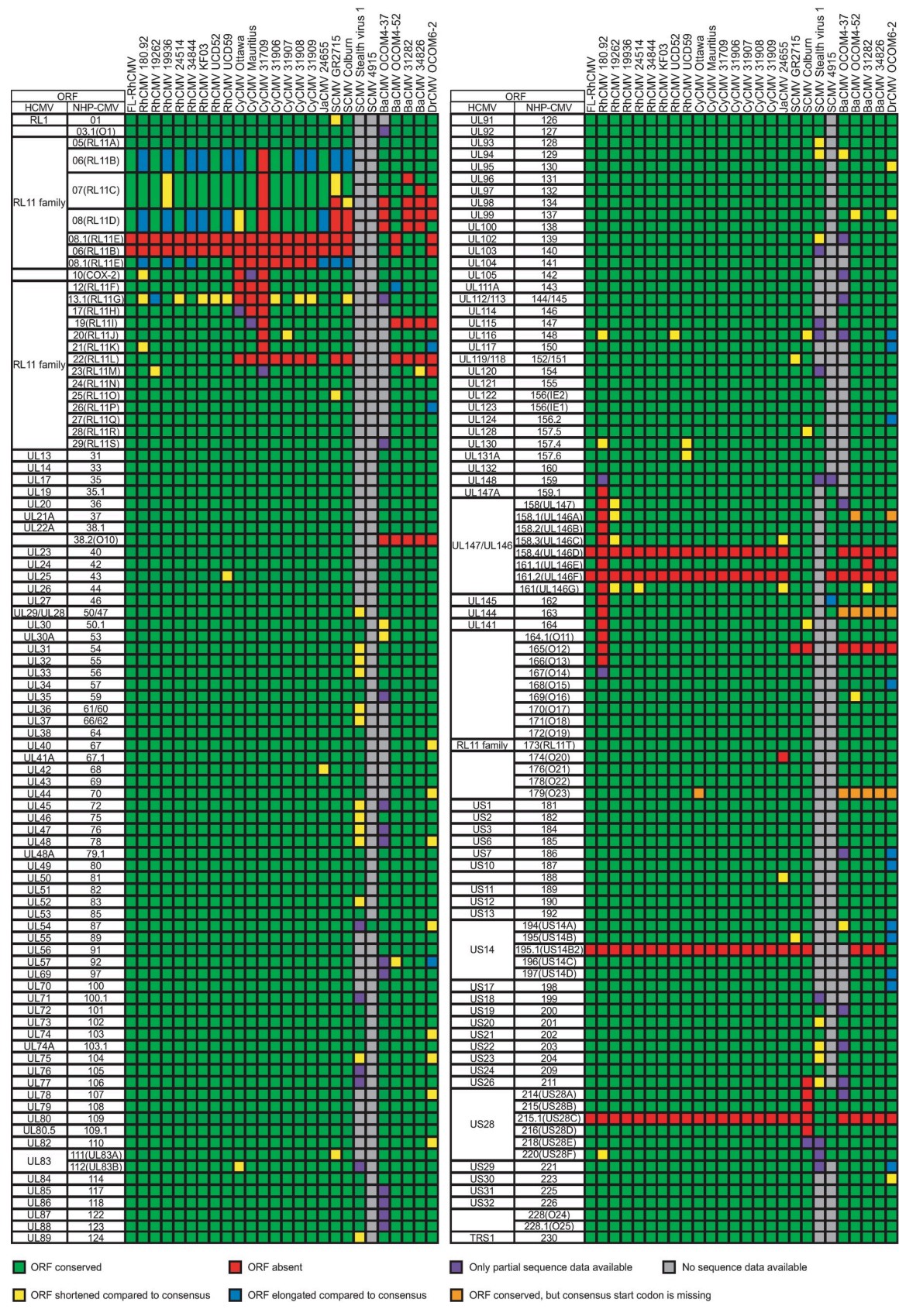

**Fig 3. Viral ORFs contained in FL-RhCMV compared to other NHP CMVs.** Full genome annotations of all listed old world NHP CMVs are shown. The leftmost column indicates the HCMV nomenclature for CMV encoded genes. Each ORFs that has a defined orthologue in HCMV and old world NHP CMVs is marked. If an orthologue cannot be clearly identified, the homologous gene family is given. The second column identifies the old world NHP CMV nomenclature. The same ORF nomenclature is used across all shown species, with the first or the first two letters corresponding to the host species (e.g. Rh for rhesus macaque). The virus strain analyzed is indicated. Green boxes indicate ORFs present in a particular strain, whereas red boxes indicate ORFs that are absent. Frameshifts or point mutations leading to shortened or elongated ORFs are highlighted in yellow or blue, respectively. Grey boxes indicate absent ORFs due to missing genome sequence information whereas ORFs with partial sequences are highlighted in purple. Orthologous ORFs that lack a conserved canonical start codon in some strains are highlighted in orange.

spread and increased cell free virus in the supernatant of infected cells suggesting that RL13-deficient HCMV mutants have a substantial growth advantage *in vitro* [49]. This interpretation is supported by the finding that RL13 remains intact when cell free spread of a clinical HCMV isolate was prevented by adding CMV-hyperimmunoglobulin (HIG) to the culture medium [50]. Furthermore, loss of RL13 could be prevented in HCMV by conditional expression of RL13 mRNA under the control of a tet operator (tetO) and growth in tet-repressor (tetR)-expressing fibroblasts [49]. As we have repaired this ORF and likely restored its function during the construction of FL-RhCMV we wanted to examine whether conditional expression of Rh13.1 would similarly affect the spread of FL-RhCMV in tissue culture. Hence, we inserted tandem tetO sequences 131 bp upstream of the Rh13.1 start codon and transfected the resulting FL-RhCMV/Rh13.1/tetO BAC DNA into telomerized rhesus fibroblasts (TRFs) expressing tetR [51]. The cells were overlaid to prevent cell-free spread and upon recovery of virus we measured viral plaques sizes after 18 days. As a control, we included a FL-RhCMV in which the Rh13.1 ORF had been deleted (FL-RhCMVΔRh13.1). The development of plaques was severely impeded in TRFs transfected with FL-RhCMV or FL-RhCMV/Rh13.1/tetO (**Fig 4A and 4B**). In contrast, FL-RhCMVΔRh13.1 spread rapidly in TRF and expression of the tetR led to a partial rescue of plaque formation by FL-RhCMV/Rh13.1/tetO (**Fig 4A and 4B**).

As an alternative approach to conditionally express Rh13.1 we explored the use of aptazyme riboswitches mediating the tetracycline dependent degradation of mRNAs *in cis* [52]. We inserted the Tc40 aptazyme sequence upstream and the Tc45 aptazyme sequence downstream of the Rh13.1 coding region in FL-RhCMV and monitored the stability of Rh13.1 and the surrounding genomic region by NGS upon recovery and propagation of virus in the presence or absence of tetracycline. FL-RhCMV/Rh13.1/apt grown in the absence of tetracycline displayed multiple mutations and deletions in this genomic region as early as passage 2 (**Fig 4C**). In contrast, by activating the aptazyme using tetracycline we were able to generate virus stocks that contained an intact Rh13.1 sequence (**Fig 4C**). These data are consistent with Rh13.1 being selected against in FL-RhCMV similar to selection against RL13 in HCMV because these homologous proteins impede spread in tissue culture. We further conclude that mutations in the Rh13.1 homologs found in many old world NHP CMV genomes are due to rapid tissue culture adaptations whereas the parental isolates likely contained an intact ORF. Thus, Rh13.1 and its homologs are preserved *in vivo*, but are selected against *in vitro*.

It was previously shown that repair of the PRC increased the ability of 68–1.2 RhCMV to infect epithelial and endothelial cells without affecting growth characteristics in fibroblasts [25]. Consistent with these earlier reports we observed that growth characteristics of PRC-repaired, but Rh13.1-deficient, 68–1.2 were comparable to that of 68–1 in rhesus fibroblasts with respect to kinetics and peak titers in a multistep growth curve (**Fig 5A**). Interestingly, FL-RhCMV/Rh13.1/apt also grew similar to 68–1 in fibroblasts despite Rh13.1 expression in the absence of tetracycline. Given the impact of Rh13.1 expression on viral spread observed above, this likely indicates rapid mutation of the Rh13.1 gene during multistep growth conditions. Since the PRC is important for entry into non-fibroblast cells [25] we quantified

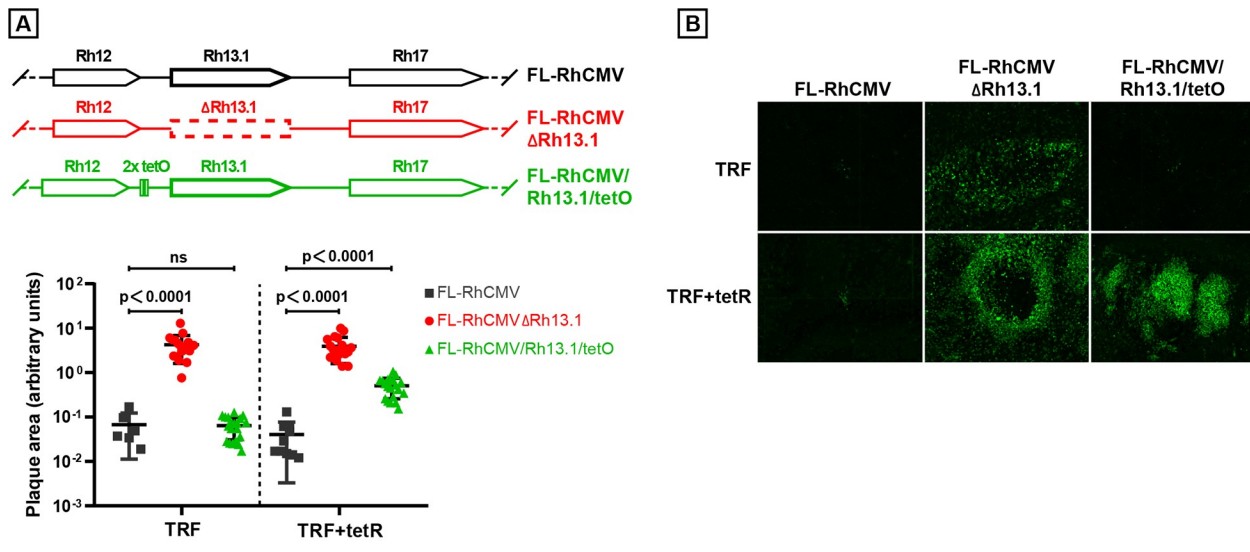

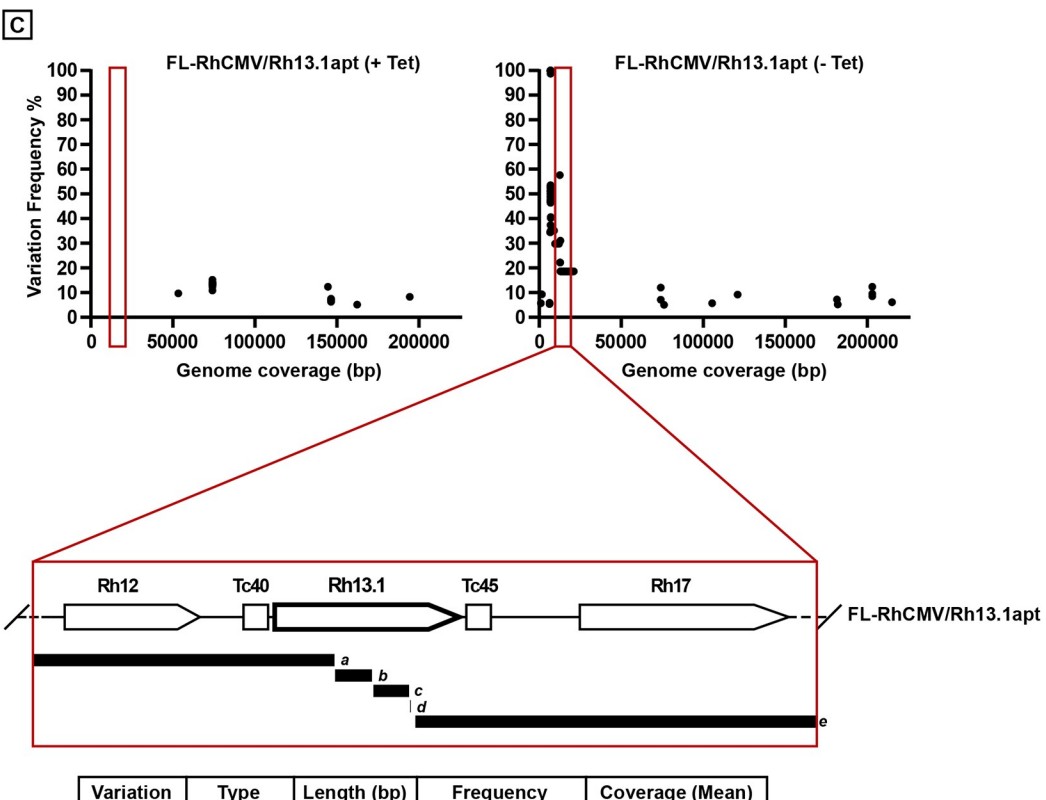

| Variation | Type | Length (bp) | Frequency | Coverage (Mean) |
|---|---|---|---|---|
| *a* | Deletion | 2877 | 23.2%-36.3% | 1239.1 |
| *b* | Deletion | 202 | 54.2%-61% | 1307.5 |
| *c* | Deletion | 174 | 20.3%-24.1% | 1239.0 |
| *d* | Insertion | 1 (T) | 31% | 1170.0 |
| *e* | Deletion | 8746 | 12.5%-24.7% | 1618.0 |

**Fig 4. Conditional expression of the RL13 homolog Rh13.1 results in reduced spreading and genomic rearrangements.** A) Deletion or reduced expression of Rh13.1 results in increased plaque size. Telomerized rhesus fibroblasts (TRF) or TRF expressing the tet-repressor (tetR) were transfected with the depicted BACs shown above. All recombinant BACs were engineered to express GFP from a P2A linker after UL36 [6]. 18 days later plaque sizes were visualized by GFP expression, and measured using ImageJ. Statistical significance was determined using an ordinary one-way ANOVA test with a p-value significance of <0.05. B) Representative images of the GFP positive plaques produced by the indicated constructs on

either TRFs or TRFs expressing the tetR are shown. C) Genetic instability of the genomic region surrounding the Rh13.1. Top: The position and relative frequencies of single nucleotide changes were determined by NGS within the genomes of FL-RhCMV/Rh13.1apt after two passages *in vitro* in the presence or absence of tetracycline. The lower panel depicts the positions of deletions/insertions of multiple nucleotides. Frequencies for each deletion are given as percentages of all reads analyzed. Since the short reads generated by the Illumina platform do not cover the entire Rh13.1 locus it is not possible to determine which deletions co-occurred in individual viral genomes resulting in combined frequencies of >100%.

infection levels of 68–1, 68–1.2 and FL-RhCMV upon entry into rhesus retinal epithelial (RPE) cells or primary rhesus fibroblasts using flow cytometry. When normalized to infected fibroblasts, 68–1 showed a strongly reduced ability to enter RPE compared to 68–1.2 (**Fig 5B**) consistent with previous reports [25]. In contrast, a FL-RhCMV vector carrying an SIVgag insert replacing Rh13.1 (FL-RhCMVΔRh13.1gag) displayed an increased ability to enter RPE cells compared to 68–1. However, infection rates on RPE cells with FL-RhCMV were consistently lower compared to 68–1.2 RhCMV in multiple independent experiments.

To independently demonstrate that the restoration of the PRC in FL-RhCMV resulted in a functional entry receptor complex, we performed entry assays into RPEs but pre-incubated the inoculum with a 26 amino acid oligopeptide representing the N-terminus of the human OR14I1 protein ("Peptide 1"). This G-protein-coupled receptors (GPCR) is located on the plasma membrane and has been shown to be required for PRC mediated HCMV entry into epithelial cells through direct interaction with the viral entry receptor [53]. As OR14I1 is highly conserved between humans and RM, pre-incubation of the inoculum with Peptide 1 should prevent PRC mediated cell entry. As demonstrated above (**Fig 5B**), PRC deficient 68–1 RhCMV was unable to efficiently enter RPEs and peptide 1 pre-incubation had no further effect, as expected. On the other hand, PRC repaired 68–1.2 RhCMV immediate early 2 (IE2) protein expression was readily detectable in RPEs after infection and pre-incubation with Peptide 1 resulted in a significant reduction of virus infected cells (**Fig 5C**). Likewise, epithelial cell infection with FL-RhCMVΔRh13.1gag was observed, but significantly inhibited by pre-incubation of the inoculum with Peptide 1 (**Fig 5C**). These results indicate that the PRC in FL-RhCMV is functional and that PRC mediated cell entry can be inhibited by pre-incubation with peptide 1. Interestingly, while FL-RhCMVΔRh13.1gag does infect RPEs, it does so less efficiently than 68–1.2, which is consistent with data presented in **Fig 5B**. In HCMV, strain specific differences in tropism can arise from alterations in the levels of both the PRC and the gH/gL/gO trimeric receptor complex which can be caused by genetic sequence variations or altered mRNA expression levels of the proteins in each complex [48,54,55]. Intriguingly, increased infection rates on epithelial cells have been reported for the PRC-repaired HCMV strain AD169 compared to PRC-intact low-passage isolates [56], a result very reminiscent of our data. This difference was determined to be due to the absence of the UL148 glycoprotein in AD169, a protein that will reduce PRC levels in favor of the trimeric gH/gL/gO complex on the virus membrane [57]. Interestingly, mRNA expression levels of the RhCMV UL148 homolog Rh159 late during infection were higher in FL-RhCMV compared to both 68–1 and 68–1.2 (**Fig 5D**). While deletion of Rh159 from the 68–1 BAC has been shown to result in significantly reduced virus replication in rhesus epithelial cells [58], this study was performed on a PRC-deficient backbone precluding any altered receptor expression levels on the cell surface, so the results probably reflect a cell entry independent mechanism. Since this gene is located within the genomic region that was inverted in 68–1, it is likely that it was put out of context of its original regulatory DNA elements, resulting in altered mRNA expression levels. Consistent with this explanation, examination of the mRNA expression levels of the late Rh137 (UL99, pp28) gene not encoded within the acquired inversion did not show any significant differences across the examined strains (**Fig 5D**). We previously demonstrated that Rh159 is an ER-resident glycoprotein that intracellularly retains NK cell activating ligands, a function that

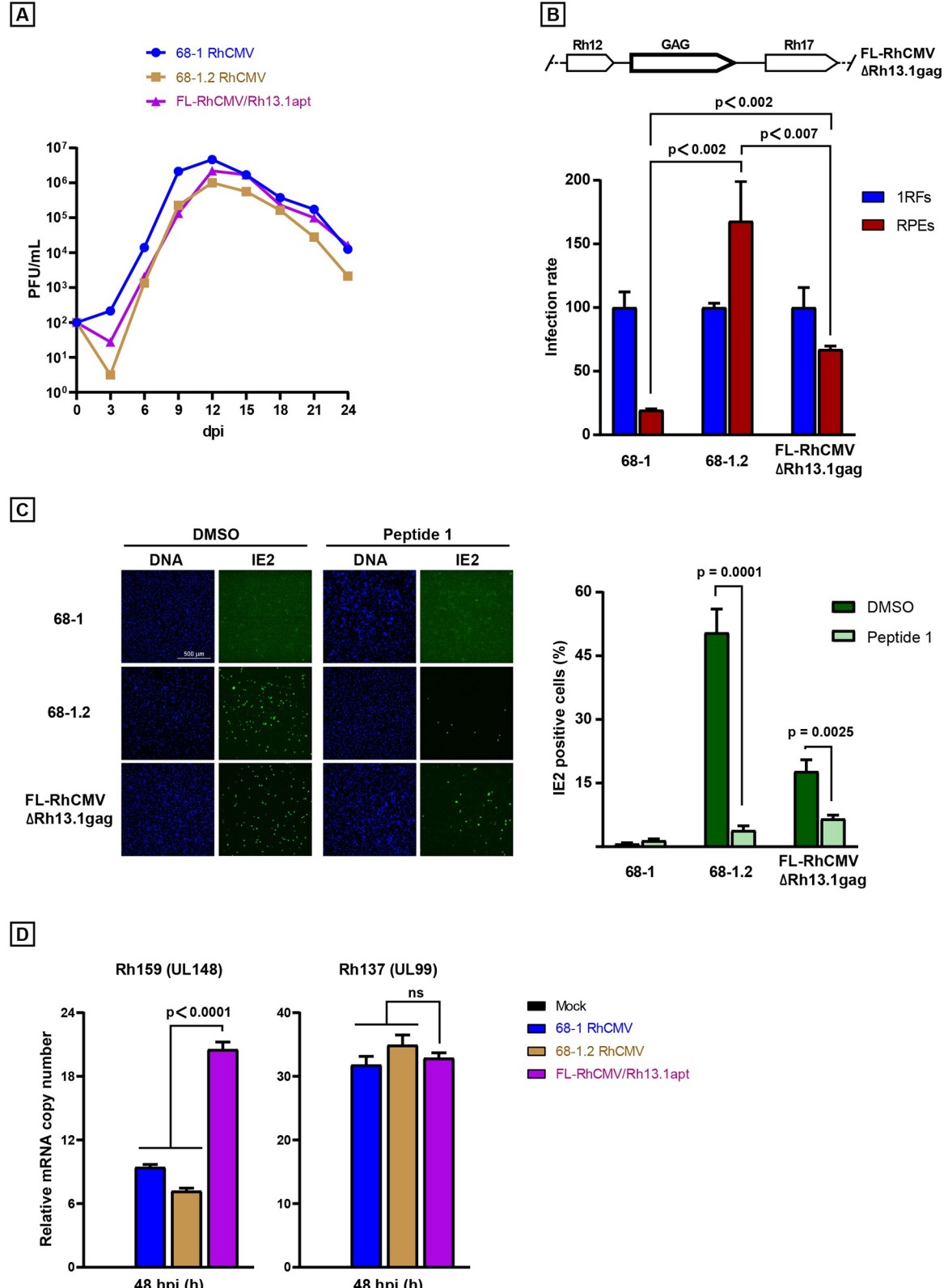

**Fig 5. Growth of FL-RhCMV *in vitro*.** A) Comparing *in vitro* growth kinetics of FL-RhCMV/Rh13.1/apt in fibroblasts to 68–1 and 68–1.2 RhCMV in a multistep growth curve. Primary rhesus fibroblasts were infected with 68–1, 68–1.2 and FL-RhCMV/Rh13.1/apt at an MOI of 0.01 on day 0. Cell culture supernatants were harvested on the indicated days and virus titers were determined by TCID50. Two

biological repeats of each sample were titrated in duplicate and the arithmetic mean of the results were graphed. B) FL-RhCMVΔRh13.1gag, depicted above the figure, shows increased infection of epithelial cells compared to 68–1 RhCMV. Primary rhesus fibroblasts or rhesus retinal epithelial cells (RPE) were infected with MOIs of 0.3 or 10, respectively, and all experiments were performed in triplicates. After 48 hours post infection, cells were harvested, fixed, permeabilized, stained with a RhCMV specific antibody [62] and analyzed by flow cytometry. Statistical significance was shown using an unpaired t-test with a p-value significance threshold of <0.05. C) Synthetic N-terminal peptide (Peptide 1) of OR14I1 blocks PRC-positive RhCMV infection of RPE cells. Virus inoculum of 68–1, 68–1.2 or FL-RhCMVΔRh13.1gag was preincubated with peptide 1 (32.6uM) or DMSO before infection of RPE cells (MOI 4.0). On day 3 pi, cells were fixed, permeabilized, immunostained for IE2 to identify infected cells, stained for DNA (blue), and imaged (4×). Results were then quantified and plotted. Data represent the mean of n = 3 experiments ±SD. Statistical significance was determined using an unpaired t-test. D) Rh159, the RhCMV homolog of UL148, is upregulated in FL-RhCMV/Rh13.1/apt. Relative mRNA copy numbers of Rh159 (UL148) and Rh137 (UL99) were determined by quantitative reverse transcription polymerase chain reaction (qRT-PCR) using specific probes. The data shown represent the mean of triplicate repeats (+/- SEM). Unpaired student t-tests with a p-value significance threshold of <0.05 were performed to show statistical significance in both graphs comparing FL-RhCMV/ Rh13.1/apt to either 68–1 RhCMV or 68–1.2 RhCMV at 48 hpi.

is not shared with UL148 [59]. However, these observations do not rule out a role of Rh159 for PRC expression and cell tropism. While further work will be required to establish this role, overall, our results indicate that FL-RhCMV is remarkably similar to low passage clinical isolates of HCMV with respect to tissue tropism and genetic stability *in vitro*.

## Kinetics and magnitude of infection by FL-RhCMV is similar to wildtype RhCMV

It was reported previously that different from the low passage isolates UCD52 and UCD59, 68–1 RhCMV displayed severely reduced viral genome copy numbers in plasma, saliva and urine in RhCMV-seronegative RM after experimental subcutaneous (s.q.) infection [26]. Concordantly, we observed significant viral genome copy numbers in all three examined bodily fluids in three female RhCMV-naïve pregnant RM infected intravenously (i.v.) with $1 \times 10^6$ pfu of RhCMV UCD52, $1 \times 10^6$ pfu of RhCMV UCD59, and $2 \times 10^6$ TCID$_{50}$ of RhCMV strain 180.92 (**Fig 6A, upper panel**) consistent with previous reports [11]. Using a qPCR primer/probe set targeting exon 1 of the IE locus, viral genome copy numbers of approximately $10^5$ copies/ml plasma were detected in all three animals between days 7 to 21, declining thereafter, whereas viral genome copy number peaked substantially later in saliva (day 42 to day 56) and urine (around day 56) reaching genome copy numbers between $10^3$ to more than $10^6$ genome copies per µg DNA in individual RM. Similar kinetics of infection and peak viremia were measured in three male RhCMV-naïve RM infected i.v. with Rh13.1-intact FL-RhCMV/Rh13.1/apt grown in the presence of tetracycline to maintain genome integrity during virus stock production (**Fig 6A, middle panel**). Since both experiments showed essentially the same development and progression of viremia in plasma, urine and saliva after inoculation (**Fig 6A, lower panel**), we conclude that *in vivo* replication of FL-RhCMV is comparable to that of low passage RhCMV isolates.

To examine whether the FL-RhCMV genome is stable *in vivo* we quantified the copy numbers of Rh13.1 (RL13) and Rh157.5 (UL128) genes by qPCR using primer/probe sets specifically targeting these genomic loci. Since viral genome copy number were virtually identical to copy numbers measured targeting exon 1 of the IE locus (**Fig 6B**), we conclude that these genes were stably maintained over the 65 days of infection monitored here. To further investigate the *in vivo* stability of all genomic region altered during the construction of FL-RhCMV, we designed primers targeting genomic DNA flanking all engineered repairs described in Fig 1 (**S2 Table**). The amplified PCR fragments were analyzed by next generation sequencing on an Illumina iSeq platform and the resulting reads were aligned to the FL-RhCMV/Rh13.1apt BAC sequence by reference guided assembly resulting in upwards of 5,000 to 60,000-fold

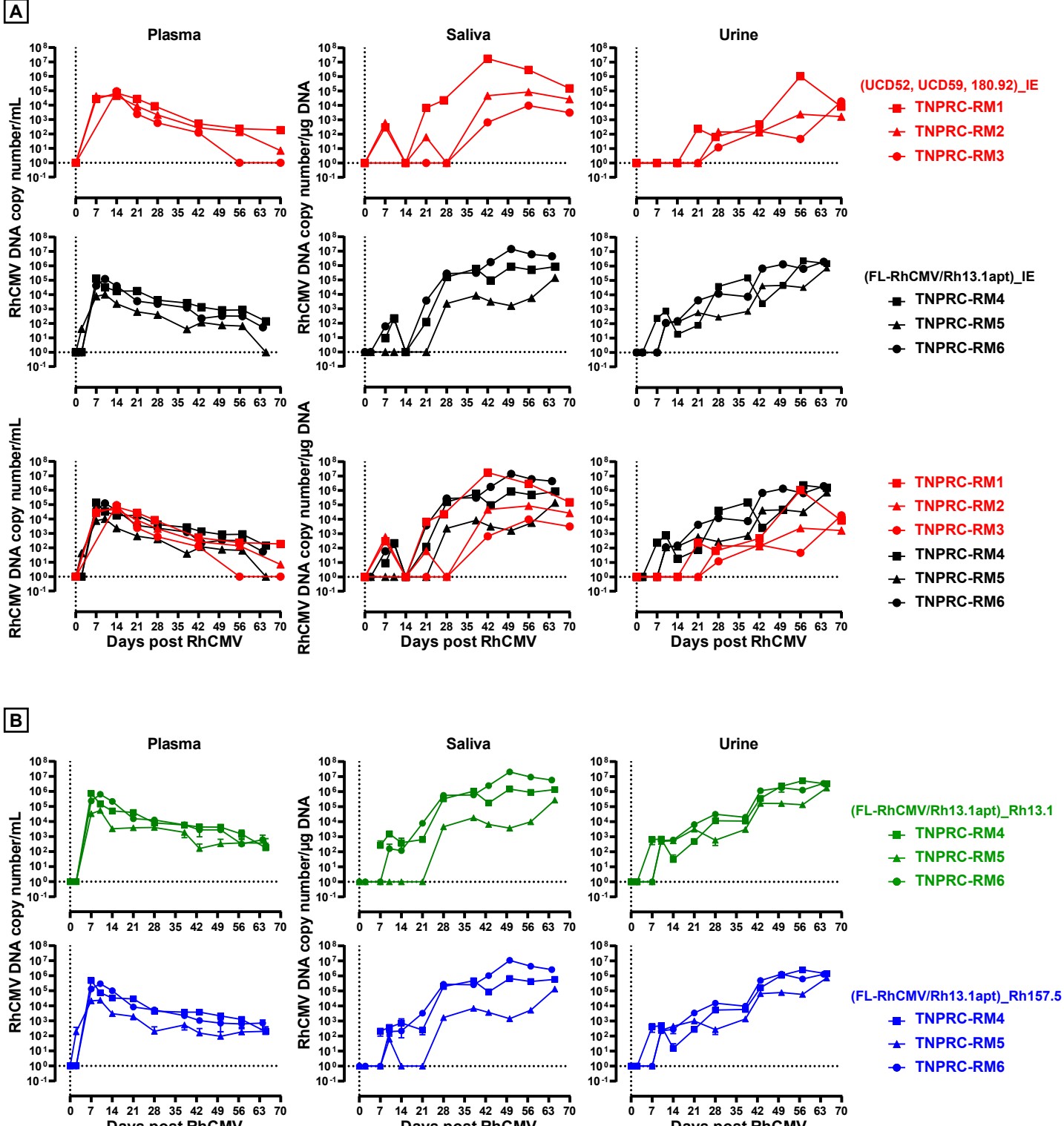

**Fig 6. In vivo replication of FL-RhCMV is similar to low passage isolates in RhCMV negative animals.** A) *Upper panel*: Replication of RhCMV isolates UCD52, UCD59 and 180.92 in RhCMV seronegative RM. Plasma, saliva, and urine RhCMV DNA loads in three RhCMV-seronegative pregnant female RM inoculated i.v. with 2 x $10^6$ TCID50 RhCMV 180.92, 1x$10^6$ PFU RhCMV UCD52, and 1x$10^6$ PFU RhCMV UCD59 are shown. The viral loads (VL) in each RM were determined at the indicated time points by qPCR targeting exon 1 of the immediate early gene (IE) region as described previously [11,84]. *Middle panel*: Replication of FL-RhCMV/Rh13.1apt in RhCMV-seronegative RM. 1.79x$10^6$ PFU of FL-RhCMV/Rh13.1apt were inoculated *i.v.* into three RhCMV-seronegative male RM and the VL was

determined on the indicated days by qPCR using the same primer/probe set as described above. *Lower panel*: The VL for all animals included in the previous panels (low passage RhCMV isolates in red and FL-RhCMV/Rh13.1apt in black) are shown in direct comparison. The data indicate that FL-RhCMV/Rh13.1apt can induce a VL comparable to commonly used virulent RhCMV isolates. B) VL in plasma, saliva, and urine were determined in the same animals shown in the middle panel of A) using qPCR primer/probe sets specific for Rh13.1 (upper panel) or Rh157.5 (lower panel) shown in green and in blue, respectively. The data indicates the presence of both genes in FL-RhCMV/Rh13.1apt at 65 days post infection while both genes are rapidly selected against during *in vitro* tissue culture on fibroblasts.

coverage (**S8 Fig**). For six of the eight fragments, no alterations compared to the clonal parental BAC were detectable. However, analysis in the Rh13.1 region indicated that 2.9% of the genomes in the inoculum used for infection of the RM had lost an 827bp segment at the end of the ORF including the introduced 3' TC45 aptazyme. Intriguingly, this deletion cannot be detected in the urine of any of the three examined animals at day 64 or 65 post inoculation, possibly indicating that the unaltered Rh13.1 was positively selected for *in vivo*. Similarly, we detected a large deletion spanning from ORF Rh59 (UL35) through ORF Rh61/Rh60 (UL36) to ORF Rh66/Rh62 (UL37) in 3.4% of the genomes contained in the inoculum (**S8 Fig**) and this deletion was not detected in any of the urine samples. However, in the urine sample of TNPRC-RM4 we detected a different, smaller deletion of this gene region only affecting ORF Rh61/Rh60 (UL36) and ORF Rh66/Rh62 (UL37) in 3% of the genomes which could indicate an independent mutation that arose *in vivo*. This observation is surprising as data examining the homologous M36 or M37 proteins in MCMV indicate that deletion of either ORF will result in significant attenuation *in vivo* [60,61]. Further studies will be required to determine whether such deletions also occur in circulating RhCMV strains. In addition, TNPRC-RM6 showed the presence of a synonymous mutation in ORF Rh66/Rh62 (UL37) in 14.6% of the genomes in urine at day 64. This silent mutation likely reflects a single nucleotide polymorphisms (SNPs) and it is presently not clear whether there is any selective advantage of this SNP or whether this change represents a random genetic drift that accumulated in this animal. Taken together, our results show that FL-RhCMV can replicate with similar kinetics and peak titers compared to known low passage RhCMV isolates. Furthermore, while the genome does show some instability in tissue culture, which is expected for a primary CMV isolate, it appears to be very stable *in vivo* in CMV naïve RM, at least for the initial 65 days of infection.

### *In vivo* dissemination of FL-RhCMV-derived viral vectors

A central goal of our research is to use the RhCMV animal model for the development of CMV-based vaccine vectors [12]. We recently reported that deletion of the pp71-encoding RhCMV gene Rh110 resulted in reduced dissemination and lack of shedding of 68-1-derived vaccine vectors [20]. Nevertheless, SIV-antigen expressing vaccines based on these live-attenuated vector backbones maintained the ability to control highly virulent SIV$_{mac239}$ upon challenge [19]. However, since 68–1 lost its homologs of the PRC subunits UL128 and UL130 as well as homologs of the viral UL146 family of CXC chemokines it was conceivable that these deletions contributed to viral attenuation. To determine the impact of these gene deletions on viral dissemination we generated viral vectors based on either FL-RhCMV or FL-RhCMV lacking the homologs of UL128, UL130 and all members of the UL146 family. As PCR- and immunological marker we selected a fusion protein of six *Mtb* antigens that was recently used to demonstrate protection against *Mtb* challenge with 68-1-based vaccine vectors [14]. As antigen-insertion site we replaced the Rh13.1 gene, thus increasing vector stability *in vitro* while using the Rh13.1 promoter to drive antigen expression.

We previously reported *in vivo* dissemination of 68–1 RhCMV in RhCMV-seronegative animals [62]. Hence we assigned six CMV-naïve RM and inoculated three with FL-RhCMVΔRh13.1/TB6Ag and three with FL-RhCMVΔRh13.1/TB6AgΔRh157.4–5ΔRh158-

161. Subsequently, we took the animals to necropsy at 14 dpi to systematically measure viral genome copy numbers in tissue samples using nested PCR as described previously [20,62]. While both recombinants resulted in significant viral accumulation at the injection sites and the nearest draining lymph node, FL-RhCMV genomic DNA was highly abundant in many of the tissues examined (**Fig 7A**). In contrast, FL-RhCMV lacking the UL128, UL130 and UL146 homologs displayed significantly reduced spreading beyond the initial site of replication (**Fig 7A**). Solely tissue samples from the spleen retained notable viral copy numbers for the deletion mutant, although at significantly reduced levels compared to FL-RhCMV (**Fig 7B and 7C**), whereas dissemination to and replication in most other tissues was almost completely abrogated. The results obtained with FL-RhCMVΔRh13.1/TB6AgΔRh157.4–5ΔRh158-161 and FL-RhCMVΔRh13.1/TB6Ag are consistent with previous observations for 68–1 RhCMV [20,62] and UCD52 and UCD59 [26], respectively. These data also suggest that RhCMV vectors with 68-1-like configuration are attenuated *in vivo*.

To determine the cell types infected *in vivo* by FL-RhCMV during primary lytic replication, we performed RNAscope *in situ* hybridization (ISH) in combination with immunohistochemistry for cellular markers on spleen tissue obtained from the same animals. Consistent with the high genome copy numbers observed, FL-RhCMVΔRh13.1TB6Ag was rapidly detected in tissue sections of the spleen, where the large clusters of infected cells primarily localized within the white pulp with fewer individual cells infected within the red pulp (**Fig 7B and 7C**). In contrast, the deletion mutant could only be detected very sparsely in a few infected cells across the examined tissue, localized primarily within the white pulp with sporadic rare viral RNA positive cells found within the red pulp. Co-staining with cellular markers identified the infected cells as vimentin-positive mesenchymal cells such as fibroblasts, whereas we did not find RhCMV RNA in CD34[+] hematopoetic stem cells or CD68/CD163-positive macrophages commonly associated with latent CMV infections. While this does not exclude the possibility that FL-RhCMV can infect other cells types, it indicates that the vast majority of cells infected in the spleen during the initial acute viremia after infection of naïve RM are in the connective tissue. Nevertheless, while the viral loads of the two different vectors differ substantially across naïve RM, they both elicited similar frequencies of TB6Ag-specific CD4[+] and CD8[+] T-cell responses in all examined tissues (**Fig 7D**). This observation is consistent with previously reported findings that, above a given threshold, T cell responses are largely independent of viral replication *in vivo* and with the reported immunogenicity of 68-1-based vectors [14,15,20,63].

## RNAseq analysis of *in vivo* tissue samples identifies multiple viral transcript that are highly expressed across tissues

The high genome copy numbers measured in several tissues of FL-RhCMVΔRh13.1/TB6Ag-inoculated RM at 14dpi provided an opportunity to compare viral gene expression from a fully characterized viral genome by RNAseq analysis *in vivo* and *in vitro*. In particular, this analysis would allow us to simultaneously verify expression of all genes encoded in the repaired loci. Total RNA was isolated from the lung, the axillary lymph node (ALN), the parotid salivary gland (PSG) and the submandibular salivary gland (SSG) as these samples showed high viral genome copy numbers (**Fig 7A**). For comparison, we infected primary rhesus fibroblast at an MOI of 5 with FL-RhCMVΔRh13.1/TB6Ag and harvested total mRNA at 8, 24 and 72 hpi representing immediate early, early and late times post infection. While the average number of reads/sample were comparable between the *in vitro* (average of 86,013,721) and *in vivo* (average of 107,502,852) RNA samples, the ratio of viral/host reads was much higher *in vitro*, particularly at late times of infection, an entirely expected result as a much lower number of cells are

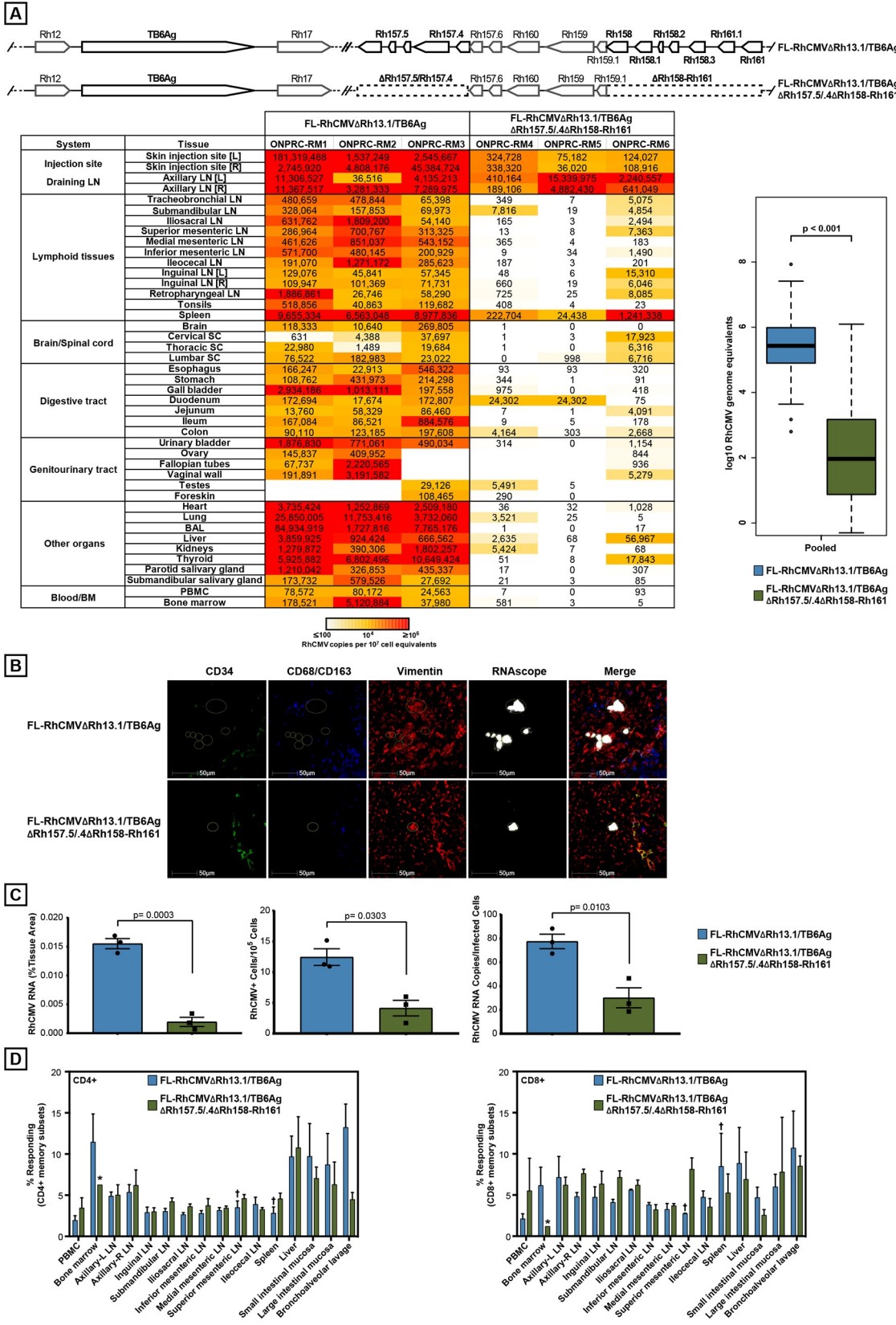

**Fig 7. Spreading of FL-RhCMV *in vivo*.** A) Tissue genome copy numbers of FL-RhCMV. Three RhCMV-naïve RM (RM1-RM3) were inoculated with $10^7$ PFU FL-RhCMVΔRh13.1/TB6Ag while another three RhCMV-naïve RM (RM4-RM6) were inoculated

with $10^7$ PFU of FL-RhCMVΔRh13.1/TB6AgΔRh157.4–5ΔRh158-161. The genome regions shown depict the alterations and deletions introduced into the FL-RhCMV backbone to create the constructs used in this experiment. All 6 RM were necropsied at day 14 post-infection and viral genome copy numbers per $10^7$ cell equivalents were determined in the indicated tissues using ultra-sensitive nested qPCR specific for TB6Ag. Statistical analysis was performed using a two-sided Wilcoxon tests (unadjusted p values < 0.05) excluding all tissues at the injection site and the nearest draining lymph nodes to detect significant differences in dissemination. B) *In situ* immunofluorescence phenotyping of cells expressing RhCMV RNA was performed by multiplexing RNAscope *in situ* hybridization with antibody detection of cellular markers specific for myeloid/macrophage cells (CD68/CD163), endothelial cells (CD34), and mesenchymal cells (vimentin) in the spleen of macaques inoculated with either FL-RhCMVΔRh13.1TB6Ag (FL-RhCMV) or FL RhCMVΔRh13.1/TB6AgΔRh157.4–5ΔRh158-161. The majority of cells inoculated with the FL-RhCMV were vimentin+ CD34- CD68/CD163-, indicating they were of mesenchymal origin. C) To quantify differences in RhCMV infection and expression levels in macaques inoculated with either FL-RhCMV or FL RhCMVΔRh13.1/TB6AgΔRh157.4–5ΔRh158-161, we used three independent quantitative approaches in the HALO image analysis platform from Indica Labs: i) the percent area of the tissue occupied by infected cells, ii) the number of infected cells per $10^5$ cells, and iii) an estimate of RhCMV viral RNA copy number per infected cell. Statistical significance was calculated using an unpaired t-test. D) Tissue distribution of TB6Ag insert–specific CD4+ and CD8+ T cell responses elicited by FL-RhCMVΔRh13.1TB6Ag versus FL RhCMVΔRh13.1/TB6AgΔRh157.4–5ΔRh158-161 vectors. Flow cytometric ICS (CD69, TNF-α and/or IFN-γ readout) was used to determine the magnitude of the CD4+ and CD8+ T cell responses to peptide mixes corresponding to the six Mtb antigens contained in the TB6Ag-fusion (Ag85A, ESAT-6, Rpf A, Rpf D, Rv2626, Rv3407). Mononuclear cells were isolated from the indicated tissues from three RhCMV-naïve RMs inoculated with $10^7$ PFU FL-RhCMVΔRh13.1TB6Ag (blue bars) and three RMs inoculated with $10^7$ PFU FL-RhCMVΔRh13.1/TB6AgΔRh157.4–5ΔRh158-161 (green bars) and all RM taken to necropsy at either 14 or 15 days post infection. Response comparisons per tissue are shown as the mean + SEM percentage of T cells specifically responding to the total of all peptide mixes (background subtracted) within the memory CD4+ or CD8+ T cell compartment for each tissue (n = 3 per tissue, unless otherwise noted by * n = 1 or † n = 2).

infected in our *in vivo* samples (S9 Fig). The absolute number of reads aligning to the annotated RhCMV ORFs for all *in vitro* and *in vivo* samples can be found in the S3 Table. Analysis of *in vitro* samples across the entire FL-RhCMV genome indicated that mRNA expression of all re-introduced genes in the repaired U$_L$-region was detectable at all time points, indicating successful restoration of a WT-like gene expression cascade (S10 Fig). Principal component analysis (PCA) on the normalized count matrix of RhCMV transcripts revealed that while the early *in vitro* samples clustered together, the late samples showed an mRNA expression pattern closer to expression profiles obtained from lung and ALN samples (Fig 8A). This is consistent with active viral replication in these tissues at this time point. In contrast, PCA revealed that gene expression profiles of PSG and SSG samples were distinct from the other tissue samples and one another. Importantly, although generated from different outbred animals, viral gene expression patterns from the same tissue source were more closely related across all three RM than across different tissue samples within the same animals. This indicates that viral mRNA expression varies within infected animals depending on the examined tissue.

Since expression patterns of the same tissues across animals were comparable, we combined these samples to compare the expression levels of each ORF between tissues and *in vitro* results. Surprisingly, this analysis revealed that some of the most highly expressed ORFs found both at late times post-infection *in vitro* and in all tissues examined *in vivo*, were the soluble chemokine binding protein Rh38.1 (UL22A) as well as two CXC chemokine-like genes of the UL146 family, Rh158.2 (UL146B) and Rh158.3 (UL146C) (Fig 8B). Normalized to the ORF size, of the ten ORFs with the most sequence coverage in each sample (Fig 8C), 80.74% mapped to these three ORFs in rhesus fibroblasts at 72 hpi, 45.94% in Lung, 31.47% in ALN, 34.38% in PSG and 32.76% in SSG. While UL22A is known to be one of the most highly expressed ORFs in HCMV [64], this dominant expression of two UL146 family members had not been observed previously. Interestingly, both UL146-homologs are deleted in 68–1 and were re-inserted during our construction of FL-RhCMV (Fig 1). The abundance of viral transcripts encoding chemokine-binding and chemokine-like proteins suggests that RhCMV interference with the host's chemokine network is a major immune modulatory strategy during the acute phase of infection.

**A**

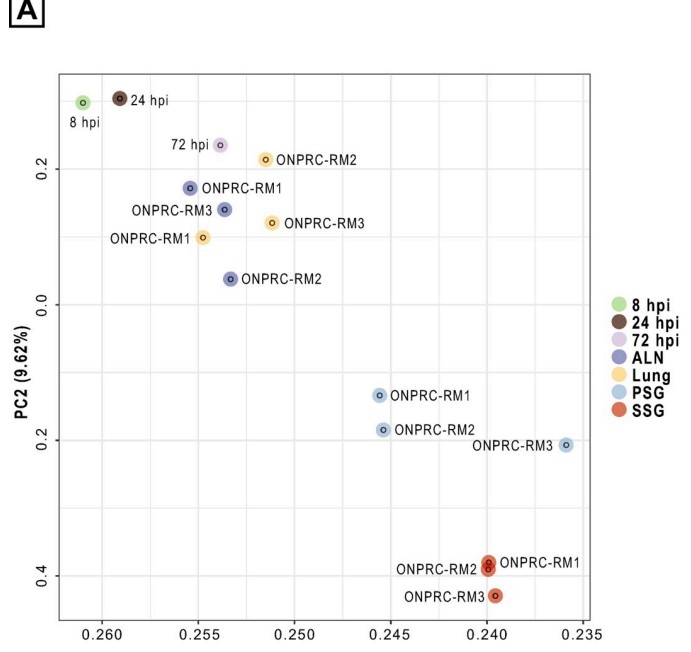

**C**

| Rhesus Fibroblasts 8 hpi | | Rhesus Fibroblasts 24 hpi | | Rhesus Fibroblasts 72 hpi | |
|---|---|---|---|---|---|
| Rh165 (O12) | 6.28% | Rh179 (O23) | 3.14% | Rh158.3 (UL146C) | 40.68% |
| Rh156 (UL123) | 4.57% | Rh181 (US1) | 2.38% | Rh158.2 (UL146B) | 30.64% |
| Rh33 (UL14) | 4.28% | Rh53 (UL30A) | 2.11% | Rh38.1 (UL22A) | 9.42% |
| Rh166 (O13) | 3.58% | Rh68 (UL42) | 2.10% | Rh160 (UL132) | 2.60% |
| Rh179 (O23) | 3.41% | Rh161 (UL146G) | 1.99% | Rh159 (UL148) | 0.88% |
| Rh160 (UL132) | 2.87% | Rh228.1 (O25) | 1.92% | Rh110 (UL82) | 0.84% |
| Rh158.3 (UL146C) | 2.84% | Rh67 (UL40) | 1.90% | Rh159.1 (UL147A) | 0.77% |
| Rh68 (UL42) | 2.38% | Rh199 (US18) | 1.89% | Rh137 (UL99) | 0.75% |
| Rh31 (UL13) | 2.25% | Rh33 (UL14) | 1.86% | Rh44 (UL26) | 0.49% |
| Rh228.1 (O25) | 2.05% | Rh160 (UL132) | 1.75% | Rh100.1 (UL71) | 0.46% |

| Axillary Lymph Node (ALN) | | Lung | | Parotid Salivary Gland (PSG) | | Submandibular Salivary Gland (SSG) | |
|---|---|---|---|---|---|---|---|
| Rh158.3 (UL146C) | 13.93% | Rh158.3 (UL146C) | 21.35% | Rh158.3 (UL146C) | 16.80% | Rh158.3 (UL146C) | 13.95% |
| Rh158.2 (UL146B) | 11.92% | Rh158.2 (UL146B) | 17.97% | Rh158.2 (UL146B) | 11.45% | Rh158.2 (UL146B) | 11.42% |
| Rh38.1 (UL22A) | 5.62% | Rh38.1 (UL22A) | 6.62% | Rh38.1 (UL22A) | 6.13% | Rh38.1 (UL22A) | 7.39% |
| Rh160 (UL132) | 2.48% | Rh35 (UL17) | 2.42% | Rh163 (UL144) | 2.54% | Rh163 (UL144) | 3.45% |
| Rh67 (UL40) | 2.42% | Rh160 (UL132) | 1.78% | Rh33 (UL14) | 1.85% | Rh103.1 (UL74A) | 2.77% |
| Rh137 (UL99) | 2.08% | Rh172 (O19) | 1.60% | Rh35 (UL17) | 1.67% | Rh67 (UL40) | 2.09% |
| Rh53 (UL30A) | 2.02% | Rh161 (UL146G) | 1.57% | Rh172 (O19) | 1.63% | Rh53 (UL30A) | 1.79% |
| Rh79.1 (UL48A) | 1.92% | Rh137 (UL99) | 1.35% | Rh161 (UL146G) | 1.46% | Rh160 (UL132) | 1.67% |
| Rh110 (UL82) | 1.85% | Rh53 (UL30A) | 1.22% | Rh79.1 (UL48A) | 1.36% | Rh67.1 (UL41A) | 1.63% |
| Rh67.1 (UL41A) | 1.44% | Rh19 (RL11I) | 1.15% | Rh143 (UL111A) | 1.30% | Rh33 (UL14) | 1.62% |

**B**

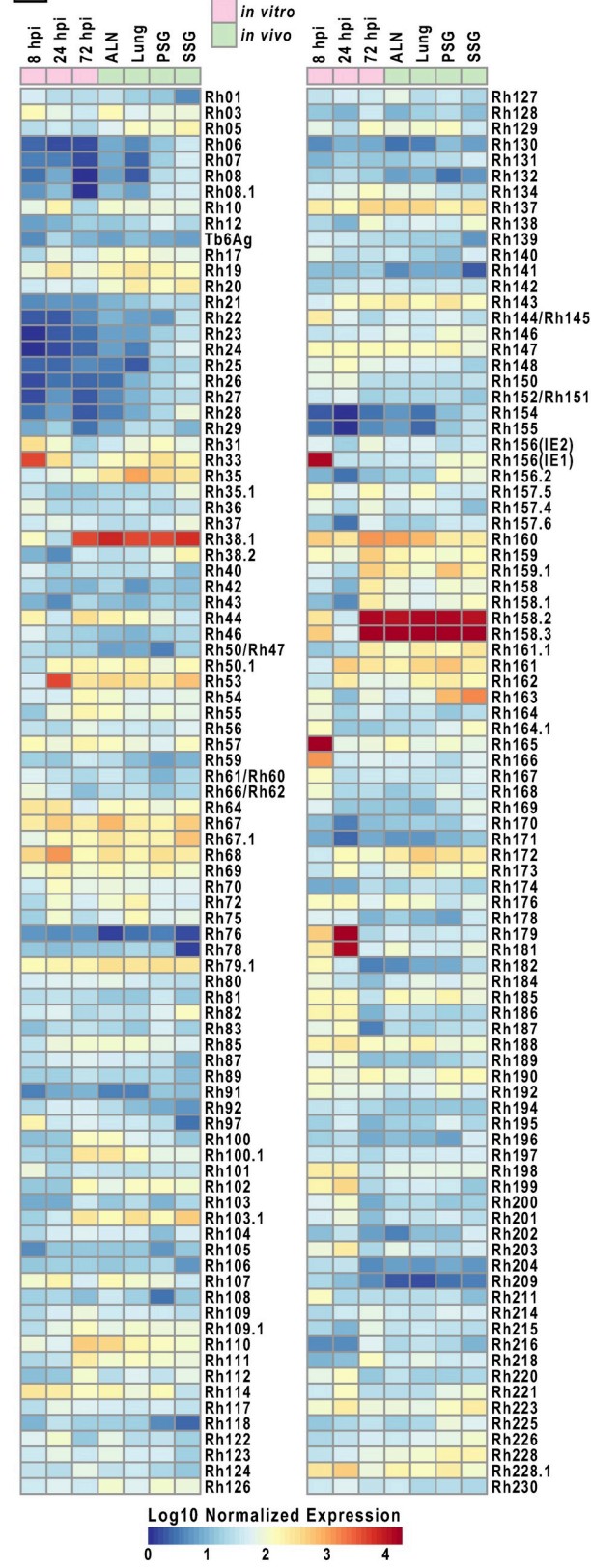

**Fig 8. Viral gene expression profile of FL-RhCMV *in vitro* and *in vivo*.** A) Comparison of *in vitro* and *in vivo* gene expression profiles by principal component analysis. *In vitro*: Rhesus fibroblasts were infected with an MOI = 5 of FLRhCMVΔRh13.1/TB6Ag and the cells were harvested at the indicated times. *In vivo*: RNA-was isolated from indicated tissues of RM1-RM3 described in Fig 7. Total RNA was isolated from all samples and RNAseq was performed on libraries build from polyA-fractionated RNA using an Illumina HiSeq-2500 next generation sequencer. PCA was done on the combined and quantile normalized expression matrix (see Materials and Methods). We observed that PC1 and PC2, shown herein, combined capture over 70% of total variance with distinct sets of co-regulated genes. B) *In vitro* and *in vivo* expression levels of each ORF. Expression levels were normalized between the *in vitro* and *in vivo* samples using quantile normalization (see Materials and Methods). C) For all samples analyzed in B) the ten viral ORFs showing the highest mRNA coverage after normalization for ORF size are shown. All values are given as percent of total viral reads mapping to all annotated ORFs normalized for size.

## Discussion

To generate a RhCMV clone that is representative of a low passage isolate we chose to repair an existing BAC clone instead of cloning a new primary isolate since this would allow us to better compare results obtained with FL-RhCMV to historic data obtained with 68–1 BAC clone-derived recombinants and thus facilitate the identification of viral determinants of tissue tropism, pathogenesis and immune response programming. Using sequence information from the original primary 68–1 isolate [40,65] and next generation sequencing of multiple primary RhCMV isolates we identified and reverted all mutations that resulted in frameshifts or premature termination codons in predicted ORFs. While we cannot rule out that additional mutations, particularly in non-coding regions, occurred during tissue culture as has been observed in HCMV [48], these cannot be unequivocally distinguished from strain-specific nucleotide polymorphisms and therefore remained unchanged. Since the original isolate likely contained a multitude of molecular clones, our FL-RhCMV could be similar to a representative of a sub-population present in the 68–1 isolate. Full genome alignments of the old world NHP CMV sequences generated in this study together with sequences previously submitted to GenBank allowed us to refine the genome annotation, enabling more precise genetic engineering of FL-RhCMV derived constructs in the future. Comparative genomics revealed a close conservation of the overall ORFeome across old world NHP CMV species (**Fig 3**) while also allowing us to identify differences acquired by individual species during co-evolution with their respective host. These distinct disparities largely consist of gene duplications in only six different loci across the genome and they are reminiscent of a poxvirus adaptation strategy deployed to adapt to antiviral pressure by the immune system known as a genomic accordion [66], albeit on a significantly longer evolutionary timescale. Hence, these gene duplications could be the results of the ongoing arms race between the virus and the host immune defenses. At a minimum, these data enable us to estimate when different CMV gene families entered the NHP CMV lineage and how they adapted over millions of years of co-evolution with their primate host (**Fig 9**).

Another advantage of our chosen repair strategy was that it allowed us to recreate a complete genome at the BAC stage in the absence of the selective forces of *in vitro* tissue culture. Indeed, upon culture of BAC derived Rh13.1-intact FL RCMV in fibroblasts we observed the rapid accumulation of mutations in the gene and the region surrounding the gene. This is strikingly similar to the clinical HCMV isolate Merlin which displayed the same instability in the homologous gene RL13 when an RL13-intact BAC was used for transfection [49]. Similar observations have been reported for additional HCMV isolates [32], although a small number of passaged strains appear capable of maintaining an intact RL13 ORF [67]. RL13 seems to limit viral spread, particularly in fibroblasts [67] but the exact mechanism of this inhibition is not clear. Rh13.1 belongs to the RL11 family of single transmembrane glycoproteins present in all old world NHP CMVs, as well as great ape and HCMV, but not in CMVs of new world primates (**Fig 9**). The functional conservation of Rh13.1 and RL13 is surprising since the RL11 family is highly diverse both within a given CMV species and especially when comparing

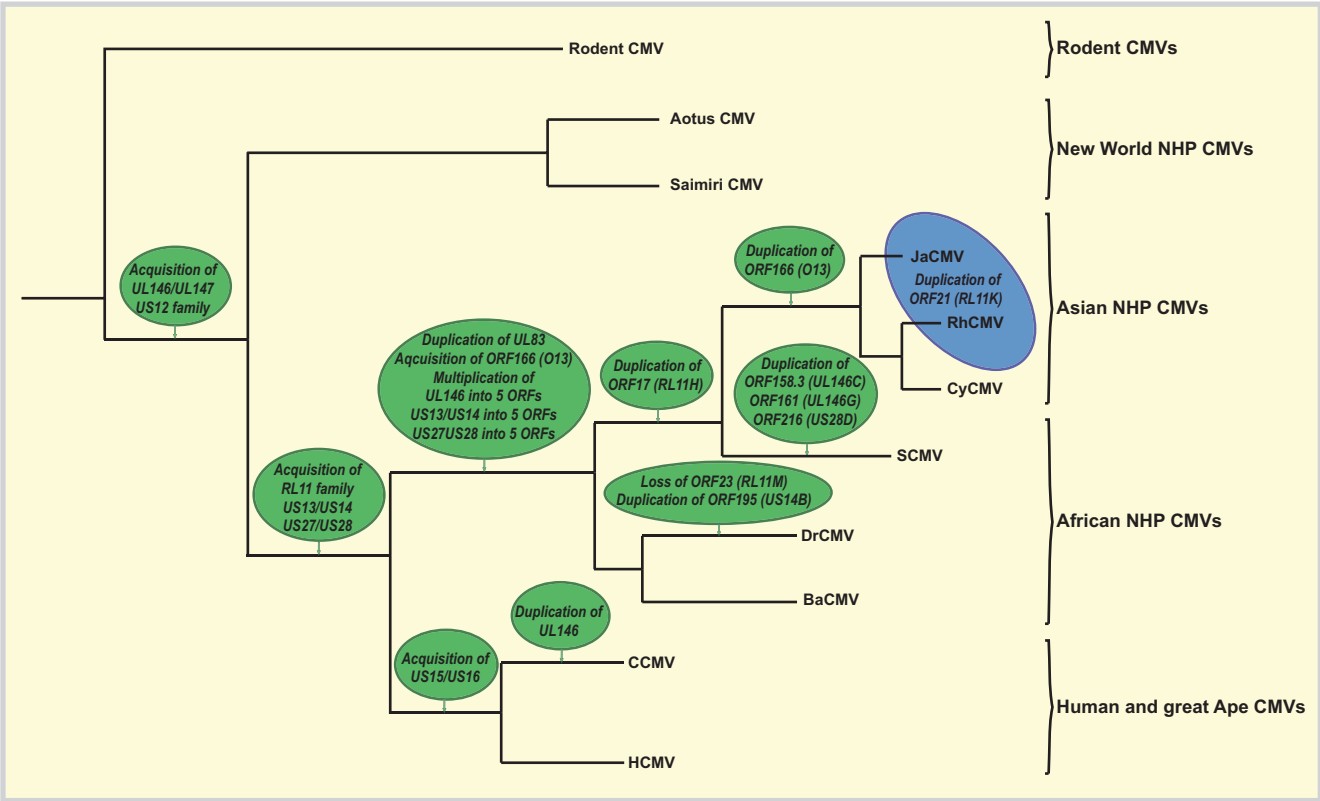

**Fig 9. Acquisition of genes and gene families in old world NHP CMVs during co-evolution with their primate hosts.** Full genome annotations of old world NHP CMVs across different species allowed for comparative genomics to identify single ORFs or entire gene families that were present in one or several CMV species but absent in others. These differences clustered in six independent loci across the genome. Examination of the phylogenetic relationship of the individual CMV species as well as their host species reveals at which point in time these gene acquisitions and gene duplications occurred. The phylogenetic tree depicted is based on the full genome alignment shown in Fig 1. The green circles indicate genetic events that took place during the evolution of each species. The blue circle represents the acquisition of RL11K, a gene duplication found in RhCMV and JaCMV but not in CyCMV. Since CyCMV appears to be more closely related to RhCMV than JaCMV by full genome alignment (see Fig 1) this appears counterintuitive. However, phylogenetic alignments of the corresponding macaque host species based on morphology [106], mitochondrial DNA data [107] or Alu elements [108] reveals that Japanese macaques (*M. fuscata*) and rhesus macaques (*M. mulatta*) speciated more recently compared to cynomolgus macaques (*M. fascicularis*).

family members between great ape and old world monkey CMVs [10,45,68]. Our data further suggest that while Rh13.1 and RL13 are selected against *in vitro*, there is a strong selection for their presence *in vivo*, since genomes carrying inactivating deletions in this gene present in the viral inoculum were absent from the population 65 days after infection of RM (**S8 Fig**). RL13 has been shown to bind to antibody Fc portions [69] and it is thus possible that it serves as an immune evasion protein in the host. Whether this function is conserved in NHP CMVs is currently unknown. To enable the study of Rh13.1-intact vectors, we therefore generated two different tetracycline-regulated systems that allow for the conditional expression of Rh13.1 so that the virus can be grown *in vitro* without selection against Rh13.1 whereas mRNA would be expressed *in vivo* in the absence of tetracycline. Indeed, we observed *in vivo* viremia of Rh13.1-intact FL-RhCMV that was comparable to low passage isolates.

However, we also observed that FL-RhCMV lacking Rh13.1 displayed substantial *in vivo* spread that was significantly more pronounced than a mutant that lacked the UL128 and UL130-homologous subunits of the pentameric complex together with all genes homologous to HCMV UL146 and UL147. These data suggest that Rh13.1-deleted viruses might maintain most of the wildtype characteristics in primary infection of immunocompetent adult animals. We also know that Rh13.1 is not required for the establishment and maintenance of persistent

infections since strain 68–1 lacks a functional Rh13.1, yet persists as shown by long-term immune responses and shedding [63]. Given a possible function of Rh13.1 in evasion of anti-body responses, it would be interesting to compare spreading of Rh13.1 intact and deleted viruses in the presence of anti-CMV antibodies. While inactivation of RL13 generally represents the first tissue culture adaptation step observed in HCMV, this is often followed by the loss of one or multiple members of the PRC [49]. Although we did not observe loss of PRC members in FL-RhCMV *in vitro*, this might be due to the limited numbers of passages we examined. Since PRC mutations occurred in 68–1 RhCMV during prolonged tissue culture it is likely that further passaging of FL-RhCMV would result in adaptations akin to HCMV indicating that the overall genome stability and the sequence of adaptation are likely similar across primate CMV species. Importantly however, all genes encoding the PRC subunit homologs of UL128-131 were stable *in vivo* (**S8 Fig**).

The strong attenuation of a "68–1 like" FL-RhCMV lacking homologs of UL128, UL130 and UL146 observed in primary infection suggests that gene deletions in these regions are the likely reason for the previously reported lack of measurable viremia and shedding of 68–1 RhCMV [26]. Further studies will be required to determine the individual contribution of pentamer subunits and UL146-related chemokine homologs to viral dissemination and spread and it is conceivable that defects in both gene regions contributed to the previously reported reduced *in vivo* tropism and decreased virus shedding in examined bodily fluids after primary infection with 68–1 compared to UCD52 and UCD59 [24,26]. In the guinea pig model it was shown that the repair of the PRC lead to a more pathogenic virus capable of congenital transmission to pubs in pregnant dams [70], likely as the result of increased systemic viral loads during the course of infection. It is furthermore possible that a PRC-independent function of the UL128 and UL130 homologs Rh157.5 and Rh157.4 could play an important role during the early stages of infection. The N-termini of both proteins show homology to host CC and CXC chemokines respectively and chemokine function has been shown for HCMV UL128 in *in vitro* chemotaxis assays [71]. Additionally, we observed that vaccine vectors based on 68–1 elicited CD8$^+$ T cell responses that were unconventionally restricted by MHC-II and MHC-E instead of classical MHC-Ia alleles whereas repair of Rh157.5 and Rh157.4 in 68–1.2 RhCMV resulted in the induction of conventionally MHC-I-restricted CD8$^+$ T cells [14,17,18]. Conceivably, the unconventionally restricted CD8$^+$ T-cell responses induced by 68-1-like RhCMV might be more efficient in controlling virus replication *in vivo* compared to conventionally restricted CD8$^+$ T cells elicited by wildtype RhCMV. Furthermore, the finding that two of the most highly expressed transcripts, both *in vitro* and *in vivo*, belong to the UL146 family, with both genes being deleted in strain 68–1, could emphasize an important role of their gene products in viral infection. The UL146 gene family of chemokine like proteins is only found in primate CMVs (**Fig 9**). Given their homology to chemokines such as interleukin 8 (CXCL8) they were likely acquired from the host. HCMV contains two family members: UL146 and UL147. While a single UL147 homologous gene can be found across all primate CMV species with a moderate level of conservation, the UL146-homologs are highly diverse within a CMV species. Moreover, while HCMVs only contain a single UL146 member, the number of genes can vary greatly in other primate CMVs (**S4 Fig**). CCMV contains two genes almost equally related to the HCMV UL146 member. However, new world NHP CMVs contain a single UL146 family member that is highly divergent. Conversely, old world NHP CMVs encode five to seven UL146 homologs. Since 68–1 lacks these highly expressed genes, they are not required for the establishment and maintenance of persistent infection but it is possible that these chemokine homologs support viremia and dissemination during primary infection or upon re-infection, a possibility reinforced by the recent observation that inserting the HCMV UL146 protein into MCMV significantly enhances virus dissemination kinetics in infected mice [72].

The *in vitro* and *in vivo* characteristics of FL-RhCMV described here are consistent with this virus being representative of wildtype RhCMV. Based on these observations we anticipate that this recombinant will be useful for RM models of CMV pathogenesis, such as the fetal inoculation model as well as a model of congenital infection [73–75]. As FL-RhCMV is shed in all examined bodily fluids to levels likely equivalent to circulating isolates, this furthermore now enables horizontal transmission studies between co-housed animals to refine the development of an effective and safe, non-transmissible vaccine vector backbone through the introduction of targeted deletions. Up to now these models relied on low passage RhCMV isolates such as UCD52 and UCD59 which are non-clonal and are difficult to modify genetically. The availability of a true wildtype clone that is amenable to BAC recombineering will help to define viral determinants of congenital infection as well as help unravel viral immune evasion strategies that limit the efficacy of vaccines or hyperimmune globulin (HIG) treatment [27] to prevent congenital infection or pathogenesis in immunocompromised hosts.

FL-RhCMV can also serve as a translational model for the development of live-attenuated vectors derived from clinical isolates of HCMV. As recently reported, our strategy for HCMV-based vectors is to start with a clinical isolate to ensure persistence and then introduce genetic modifications that increase vector safety while maintaining desired immunological features [34]. The availability of a complete RhCMV genome will allow us to recapitulate HCMV-vector design strategies and test these designs in RM challenge models for AIDS, tuberculosis and malaria. FL-RhCMV-based vectors will thus be highly useful for both basic and translational aspects of CMV research.

## Materials and methods

### Ethics statement

All RMs housed at the ONPRC and the TNPRC were handled in accordance with good animal practice, as defined by relevant national and/or local animal welfare bodies. The RMs were housed in Animal Biosafety level (ABSL)-2 rooms. The rooms had autonomously controlled temperature, humidity, and lighting. Study RM were both pair and single cage-housed. Regardless of their pairing, all animals had visual, auditory and olfactory contact with other animals within the room in which they were housed. Single cage-housed RMs received an enhanced enrichment plan and were overseen by nonhuman primate behavior specialists. Animals were only paired with one another if they were from the same vaccination group. RMs were fed commercially prepared primate chow twice daily and received supplemental fresh fruit or vegetables daily. Fresh, potable water was provided via automatic water systems. The use of nonhuman primates was approved by the ONPRC and the TNPCR Institutional Animal Care and Use Committees (IACUC). Both institutions are Category I facilities and they are fully accredited by the Assessment and Accreditation of Laboratory Animal Care International and have an approved Assurance (#A3304-01) for the care and use of animals on file with the NIH Office for Protection from Research Risks. The IACUCs adhere to national guidelines established in the Animal Welfare Act (7 U.S.C. Sections 2131–2159) and the Guide for the Care and Use of Laboratory Animals (8th Edition) as mandated by the U.S. Public Health Service Policy. For BAL procedures, monkeys were restrained and sedated by intramuscular injection of ketamine (~7 mg/kg) with dexmedetomidine (~15 μg/kg). Following the procedure (and blood collection), atipamezole (5 mg/mL; the dose volume was matched to that of dexmedetomidine) was administered by intramuscular injection to reverse the effects of the dexmedetomine sedation. To prepare RMs for blood collection alone, monkeys were administered ketamine only as described above. Monkeys were bled by venipuncture (from the femoral or saphenous veins) and blood was collected using Vacutainers. Monkeys were humanely

euthanized by the veterinary staff at ONPRC and TNPRC in accordance with end point policies. Euthanasia was conducted under anesthesia with ketamine, followed by overdose with sodium pentobarbital. This method is consistent with the recommendation of the American Veterinary Medical Association.

## Cells and viruses

Telomerized rhesus fibroblasts (TRFs) have been described before [76]. Primary embryonal rhesus fibroblasts were generated at the Oregon National Primate Research Center (ONPRC). Both cell lines were maintained in Dulbecco's modified Eagle's medium (DMEM) with 10% fetal bovine serum and antibiotics (1× Pen/Strep, Gibco), and grown at 37˚C in humidified air with 5% $CO_2$. Rhesus retinal pigment epithelial (RPE) cells were a kind gift from Dr. Thomas Shenk (Princeton University, USA) and were propagated in a 1:1 mixture of DMEM and Ham's F12 nutrient mixture with 5% FBS, 1 mM sodium pyruvate, and nonessential amino acids. Monkey kidney epithelial (MKE) [77] cells were maintained in DMEM-F-12 medium (DMEM-F12) (Invitrogen) supplemented with epithelial cell growth supplement (ScienCell), 1 mM sodium pyruvate, 25 mM HEPES, 100 U/ml penicillin, 100g/ml streptomycin, 2 mM L-glutamine (Invitrogen), and 2% fetal bovine serum/SuperSerum (Gemini Bio-Products). The 68–1 RhCMV BAC [22] has been characterized extensively. The BAC for 68–1.2 RhCMV [25] which was based on the 68–1 BAC was provided by Dr. Thomas Shenk (Princeton University, USA). Both 68–1 and 68–1.2 viruses were derived via electroporation (250V, 950μF) of BAC-DNA into primary rhesus fibroblasts. Full cytopathic effect (CPE) was observed after 7–10 days and the supernatants were used to generate viral stocks. UCD52 and UCD59 RhCMV have been continuously passaged on MKE cells to maintain their PRC and to minimize tissue culture adaptations. To generate enough viral DNA for a full genome analysis, monolayers of MKE at 90–100% confluency were inoculated with RhCMV (MOI: 0.01). Infections progressed to ~90% CPE at which time supernatant (SN) and cells were collected and centrifuged at 6000 x g for 15 minutes at 4˚C. The SN was passed through a 0.45μm filter and was subsequently centrifuged at 26,000 x g for 2 hours at 4˚C. The SN was decanted and the virus pellet was resuspended and washed in ~ 20ml cold 1X PBS. The virus was pelleted by ultracentrifugation at 72,000 x g (Rotor SW41Ti at 21,000 rpm) for 2 hours at 4˚C, which was then repeated once more. Finally, the SN was decanted and the remaining viral pellet was thoroughly resuspended in ~1–2 ml of cold 1X PBS. The Viral stock was stored in 50μl aliquots and viral DNA was isolated from these viral stocks using the QIAamp DNA Mini Kit (Qiagen).

## BAC recombineering using *en passant* homologous recombination

Recombinant RhCMV clones were generated by *en passant* mutagenesis, as previously described for HCMV [78], and adapted by us for RhCMV [79]. This technique allows the generation of "scarless" viral recombinants, i.e. without containing residual heterologous DNA sequences in the final constructs. The homologous recombination technique is based on amplifying an I-SceI homing endonuclease recognition site followed by an aminoglycoside 3-phosphotransferase gene conferring kanamycin resistance (KanR) with primers simultaneously introducing a homology region upstream and downstream of the selection marker into the intermediate BAC cloning product. As *en passant* recombinations are performed in the GS1783 *E-coli* strain that can be used to conditionally express the I-SceI homing endonuclease upon arabinose induction [78], expression of the endonuclease with simultaneous heat shock induction of the lambda (λ) phage derived Red recombination genes will lead to the induction of selective DNA double strand breaks with subsequent scarless deletion of the

selection marker. The immunologically traceable markers used in the study, namely the SIV-mac239 GAG protein as well as the *Mtb* Erdman strain derived TB6Ag fusion protein, have been described before [14,63]. To introduce these genes into the FL-RhCMV backbone, we first introduced a homology region flanking an I-SceI site and a KanR selection marker into the selected inserts. We then amplified the transgenes by PCR and recombined the entire insert into the desired location in the FL-RhCMV BAC. The KanR cassette was subsequently removed scarlessly as described above. All recombinants were initially characterized by XmaI restriction digests and Sanger sequencing across the modified genomic locus. Lastly all vectors were fully analyzed by next generation sequencing to exclude off-target mutations and to confirm full accordance of the generated with the predicted full genome sequence.

## Isolation of the old world non-human primate CMV species from urine samples

Virus isolation was performed as previously described [41]. Briefly, urine samples were obtained through collection from cage pans, by cystocentesis or following euthanasia. From samples collected at the ONPRC we isolated BaCMV 31282 from a male olive baboon (*Papio anubis*), BaCMV 34826 from a female hamadryas baboon (*Papio hamadryas*), CyCMV 31709 form a female cynomolgus macaques (*Macaca fascicularis)* of Cambodian origin, JaCMV 24655 from a male Japanese macaque (*Macaca fuscata*) and RhCMV 34844 from a male rhesus macaque (*Macaca mulatta*) of Indian origin. Additionally, we successfully isolated RhCMV KF03 from a cage pan collected urine sample from a male rhesus macaque (*Macaca mulatta*) of Indian origin housed at the Tulane National Primate Research Center (TNPRC). All urine samples were first clarified from solid contaminants by centrifugation at 2,000 x g for 10 minutes at 4˚C and then filtered through a 0.45 μm filter (Millipore) to clear the urine of any bacterial or fungal contamination. Next, we spin-inoculated 0.5 ml– 2 ml of clarified urine onto primary rhesus fibroblasts in a 6 well plate at 700 x g for 30 minutes at 25˚C. The cells were placed on a rocker for 2 hours at 37˚C and, after removing the inoculum, washed once with PBS. The infected cells were cultured in DMEM plus 10% fetal bovine serum for 2–3 days, trypsinized and seeded in a T-175 cell culture flask. All samples were monitored weekly for CPE for up to six weeks or until plaque formation was visible. Every two weeks or after the appearance of plaques, cells were trypsinized and re-seeded to facilitate viral spread through the entire monolayer. Virus propagation was kept to an absolute minimum and viral stocks were prepared with the minimum number of passages required to be able to infect eight T-175 flasks for stock production (typically 1–3 passages).

## Isolation and purification of viral DNA for next generation sequencing (NGS)

The modified Hirt extraction [80] protocol used for the preparation of CMV viral DNA has been described [41]. Briefly, supernatants from cells that were spin-inoculated with the original urine sample were collected at full CPE and used to infect three T-175 flasks of primary rhesus fibroblasts. After 7–10 days, the supernatant was harvested and clarified by centrifugation, first at 2,000 x g for 10 minutes at 4˚C and subsequently at 7,500 x g for 15 minutes. Virus was pelleted through a sorbitol cushion (20% D-sorbitol, 50 mM Tris [pH 7.4], 1 mM $MgCl_2$) by centrifugation at 64,000 x g for 1 hour at 4˚C in a Beckman SW28 rotor. The pelleted virus was resuspended in 500μl 10.1 TE Buffer (10mM Tris, pH 8.0; 0.1mM EDTA, pH 8.0) and 500 μl 2x lysis buffer (20mM Tris-Cl, pH 8.0; 50mM EDTA, pH8.0; 200mM NaCl; 1.2% w/v SDS) was added. To digest the purified virion and to release the viral DNA, 250μg Proteinase K was added and the solution was incubated for 2h at 37˚C. This was followed by two rounds

of phenol/chloroform extractions and the viral DNA was precipitated overnight with absolute ethanol at −80˚C. The DNA was pelleted for 20 minutes at 13,200 x g at 4˚C, washed once with 70% ethanol, and subsequently resuspended in autoclaved double deionized water. DNA concentrations were determined using a ND-1000 Spectrophotometer (NanoDrop Technologies, Inc.).

## Generation of next generation sequencing libraries and next generation sequencing

Illumina sequencing libraries were generated by first fragmenting the viral DNA using an S2 Sonicator and by subsequently converting the fragments into libraries using the standard Tru-Seq protocol. All libraries were quality controlled on a Bioanalyzer (Agilent) and library concentration was determined by real time PCR and SYBR Green fluorescence. Finally, the next generation sequencing was performed on either an iSeq- or a MiSeq Next-Generation Sequencing platform (Illumina). The libraries were multiplexed at equal concentrations and loaded into a reagent cartridge at 9 pM and single read sequencing was performed for 300 cycles with 6 additional cycles of index reads. The Geneious Prime software was used for all NGS data analysis. To minimize sequencing errors, the sequencing reads were trimmed of all regions exceeding the error probability limit of 0.1%. All reads with a total length of less than 50 bp after trimming were eliminated from further analysis to decrease the background due to unspecific alignment of reads during *de novo* and reference guided assemblies. All full genome sequences were first *de novo* assembled using the processed sequencing data, and subsequently all reads were aligned to the generated consensus sequence in a reference-guided assembly. All detected single nucleotide polymorphisms (SNPs) that showed a frequency of more than 50% in a location with a depth of at least 10% of the mean depth were manually curated. All nucleotide changes that were considered to be likely the results of actual changes opposed to sequencing errors or software alignment issues, were changed in the consensus sequences and the referenced guided assemblies were repeated until no SNP showed a frequency of 50% or more. The resulting final sequence was considered the representative consensus sequences of all clones contained in the primary viral isolates.

## Nucleotide sequences used and generated in this study

All full genome old world NHP CMV sequences generated in this study were submitted to GenBank. The accession numbers for the submitted isolates are as follows: BaCMV 31282 (MT157321), BaCMV 34826 (MT157322), CyCMV 31709 (MT157323), JaCMV 24655 (MT157324), RhCMV 34844 (MT157328), RhCMV KF03 (MT157329), RhCMV UCD52 (MT157330) and RhCMV UCD59 (MT157331). Furthermore, we submitted an updated annotation for the RhCMV 68–1 BAC (MT157325), a full annotation for the partially repaired RhCMV 68–1.2 BAC (MT157326) as well as a full annotation for the FL-RhCMV BAC described here (MT157327) which was based on 68–1 and 68–1.2. Genome annotations were fine-tuned utilizing these and other NHP CMV sequences that had been previously submitted to GenBank, either by us: CyCMV 31906 (KX689263), CyCMV 31907 (KX689264), CyCMV 31908 (KX689265), CyCMV 31909 (KX689266), RhCMV 180.92 (DQ120516, AAZ80589), RhCMV 19262 (KX689267), RhCMV 19936 (KX689268) and RhCMV 24514 (KX689269) or by others: CyCMV Ottawa (JN227533, AEQ32165), CyCMV Mauritius (KP796148, AKT72642), SCMV Colburn (FJ483969, AEV80601), SCMV GR2715 (FJ483968, AEV80365), SCMV Stealth virus 1 strain ATCC VR-2343 (U27469, U27238, U27770, U27627, U27883, U27471), BaCMV OCOM4-37 (AC090446), BaCMV OCOM4-52 (KR351281, AKG51610.1), DrCMV OCOM6-2 (KR297253, AKI29779). Lastly, we also used full genome sequences of

CMV species from outside the old world NHPs in our phylogenetic analysis to classify the evolutionary relationship of FL-RhCMV to other CMVs. These additional sequences include: MCMV Smith (GU305914, P27172), RCMV Maastricht (NC_002512), RCMV England (JX867617), RCMV Berlin (KP202868), GPCMV 22122 (KC503762, AGE11533), AoHV-1 S34E (FJ483970, AEV80760), SaHV-4 SqSHV (FJ483967, AEV80915), CCMV Heberling (AF480884, AAM00704), and HCMV TR3 (MN075802).

## Phylogenetic analysis of isolated NHP CMV species

Full nucleotide sequences of rodent and primate CMV genomes were aligned using ClustalW2 [81]. The multiple sequence alignment was imported into Geneious Prime and a Neighbor-joining phylogenetic tree was build using the Geneious Tree Builder application and selecting the Jukes-Cantor genetic distance model using the MCMV Smith strain as an outgroup. The validity of the tree topology obtained was tested by using bootstrap analysis with 100 resamplings from the aligned sequences, followed by distance matrix calculations and calculation of the most probable consensus tree with a support threshold of 50%.

## Conditional expression of the Rh13.1 (RL13) encoded mRNA using the Tet-Off system

TRF were engineered to express the tetracycline-repressor (tetR) as previously described [82,83]. Briefly, a retrovirus was generated from vector pMXS-IP expressing the tetR ORF. Retronectin was then used to transduce TRFs with high efficiency, before selection in puromycin (1μg/ml). Successful expression of the tetR was validated using replication deficient recombinant adenovirus vectors expressing GFP from a tetO-regulated promoter [83].

To create a vectors conditionally expressing Rh13.1, we inserted dual tetO sequences 131bp upstream of its start codon. Next, eGFP was inserted as a marker of infection. eGFP was linked to Rh60/61 (homologous to HCMV UL36) via a P2A linker. We have previously shown for HCMV that this provides early expression of GFP, without affecting UL36 function [6]. As a control, we created a virus in which we deleted the entire Rh13.1 ORF. All vectors were created using *en passant* mutagenesis.

To analyze the effects of Rh13.1 on plaque formation, BAC DNA was transfected into TRFs or tetR-expressing TRFs using an Amaxa Nucleofector (basic fibroblast kit, program T-16). The formation of plaques was then monitored by imaging for eGFP fluorescence at various timepoints, using a Zeiss Axio Observer Z1.

## Rh13.1 (RL13) mRNA regulation using riboswitches

To generate a FL-RhCMV vectors carrying riboswitches, we inserted the published Tc45 aptazyme sequence 19bp upstream and the Tc40 aptazyme sequence 17bp downstream of Rh13.1 [52] into the BAC by incorporating them into the homologous recombination primers. The entire 122bp sequences were introduced into the 5' and the 3' flanking regions of Rh13.1 in two independent recombination steps. The resulting BAC construct was analyzed by XmaI endonuclease restriction digest, Sanger sequencing and next generation sequencing.

To reconstitute the virus, we transfected BAC DNA into primary rhesus fibroblast using Lipofectamine 3000 (ThermoFischer) in the presence of 100 μM tetracycline which was replenished every other day. For comparison, we reconstituted virus in the absence of tetracycline. After minimal passaging in the presence or absence of tetracycline virus stocks were generated, viral DNA was isolated and NGS was performed.

## Multi-step growth curves of RhCMV on primary rhesus fibroblasts

Primary rhesus fibroblast were seeded in 24 well plates and infected in duplicate with RhCMV constructs at an MOI of 0.01. The inoculum was removed after 2h and 1ml DMEM complete was added. Supernatants were collected every third day for 24 days, cells and cell debris were removed by centrifugation for 2 min at 13,000 rpm and the samples were stored at -80˚C. Viral titers were determined using a fifty-percent tissue culture infective dose (TCID50) assay in two technical repeats. Final titers were calculated using the arithmetic mean of the technical and biological repeats.

## RhCMV entry assays into primary rhesus fibroblast and rhesus retinal pigment epithelial cell

Stocks for RhCMV strains 68–1, 68–1.2 and FL-RhCMV were generated and viral titers were determined by TCID50. Infection levels in fibroblasts were experimentally equalized across stocks. Primary rhesus fibroblast were infected at an MOI of 0.3 and RPEs at an MOI of 10 as these MOIs were experimentally determined to result in optimal infection levels. The cells were fixed at 48 hpi and the overall percentage of RhCMV positive cells were determined by flow cytometry using a RhCMV-specific antibody [62]. To compare infection levels between the two cell types, we set the mean infection level in fibroblasts determined in triplicate repeats to 100% and expressed the mean infection level of triplicate repeats in RPEs in relation to the infection level in fibroblasts.

## Peptide blocking assay

Peptide 1 (MDNLTKVTEFLLMEFSGIWELQVLHA) was mixed at a final concentration of 32.6 μM with virus inoculum for 68–1, 68–1.2 or FL-RhCMVΔRh13.1gag (MOI = 4.0) and incubated at 37˚C for 2 h while rocking. The viruses were then added to RPEs grown in 96-well plates. After 2h of incubation, unbound virus and peptide were removed and replaced with fresh culture medium. At 3 dpi, cells were fixed with 4% paraformaldehyde, permeabilized with 0.1% Triton X-100, and immunostained using a mouse monoclonal α-RhCMV IE2 antibody (clone 11A5.2) [41,59] followed by an Alexa Fluor 488 goat anti-mouse secondary antibody (SouthernBiotech) and stained for DNA using Hoechst 33342 (Invitrogen). Cells were imaged on an automated ImageExpress Micro microscope (Molecular Devices) at 4× magnification. Images were analyzed with MetaXpress (Molecular Devices) to determine the total number of cells and the percentage of infected cells in each well.

## Quantitative PCR (qPCR) analysis to assess mRNA expression levels

Primary rhesus fibroblasts were seeded in 6-well plates and infected either with FL-, 68–1, or 68–1.2 RhCMV at a MOI of 1. Total RNA was then isolated at 48 hours post infection (hpi). Uninfected rhesus fibroblasts were used as a negative control. After cDNA synthesis, the quantitative PCR (q-PCR) assay was performed using primers and probes specific to each gene of interest. Rh159_forward_primer: 5’ TCAGAAATGAAGGGCAATTGTG 3’. Rh159_reverse_ primer: 5’ GCGAGCTGGCGACGTT 3’. Rh159_probe: 6FAMTATCACTCGGCTATTATCM GBNFQ. Rh137_forward_primer: 5’ GGCGCAACATACTACCCAGAA 3’. Rh137_reverse_ primer: 5’ GTAGCCATCCCCATCTTCCA 3’. Rh137_probe: 6FAMCACAACTAACTC TGGCCTTMGBNFQ. GAPDH_forward_primer: 5’ TTCAACAGCGACACCCACTCT 3’. GAPDH_reverse_primer: 5’ GTGGTCGTTGAGGGCAATG 3’. GAPDH_probe: 6FAMCC ACCTTCGACGCTGGMGBNFQ. The mRNA copy numbers for Rh159 (UL148) and Rh137 (UL99) were calculated and graphed as relative mRNA copy numbers normalized to the housekeeping gene (GAPDH).

## Intravenous (i.v.) inoculation of RhCMV-naïve RM with FL-RhCMV

Three immunocompetent male RhCMV-seronegative Indian-origin RM from the expanded SPF colony at the TNPRC were intravenously inoculated with $1.79 \times 10^6$ pfu of FL-RhCMV/Rh13.1/apt. Blood, saliva and urine were collected at biweekly to weekly intervals until necropsy 9 weeks post RhCMV inoculation. Saliva samples were collected through an oral mouthwash with sterile PBS, while urine was collected from a clean cage pan catch. RhCMV PCR was performed on DNA extracted from plasma, urine and saliva as previously described [84,85]. Briefly, urine and saliva samples were spun to remove debris, supernatants were concentrated using Centriprep YM-30 centrifugal filter units (EMD Millipore, Billerica, MA), and aliquots of concentrate were stored at -20°C for subsequent DNA extraction. DNA was extracted from plasma and saliva using the QIAmp DNA mini kit (Qiagen, Valencia, CA) and from urine using the QIAmp RNA mini kit (Qiagen, Valencia, CA). For the real-time PCR of RhCMV DNA we initially used a primers/probe set (forward primer 5'-GTTTAGGGAAC CGCCATTCTG-3', reverse primer 5'-GTATCCGCGTTCCAATGCA-3', probe 5'-FAM-TC CAGCCTCCATAGCCGGGAAGG-TAMRA-3') targeting a region in exon 1 of the immediate early gene of rhesus CMV as previously described [84]. We also designed and tested additional primer/probe sets targeting Rh13.1 forward_primer: 5' CATCTGCTCCCTTCGGAGAA 3'. Rh13.1_reverse_primer: 5' CATTGACTTCACAGCGCAAGA 3'. Rh13.1_probe: 5'-6FAM-AAAAGTCCTCAATCAAC-MGBNFQ-3' and Rh157.5 (Rh157.5_forward_primer: 5' CTCAACTACCTCAGGTCAATGTGACT 3'. Rh157.5_reverse_primer: 5' CAGGCGT TGGTGGCATAGTA 3'. Rh157.5_probe: 5'-6FAM-TGCTACCTCTGCTTCAT-MGBNFQ-3') to ensure the presence of these ORFs in the viral genome during *in vivo* replication over time. The IE exon 1 specific primers/probe set was used in a 25µL reaction with Supermix Platinum Quantitative PCR SuperMix-UDG (Invitrogen). The reactions were performed in a 96-well plate for real time quantification on an Applied Biosystems 7900HT Fast Real-Time PCR System. Absolute quantification was performed using ten-fold dilutions of a plasmid standard containing the target sequence diluted in genomic DNA derived from RhCMV-seronegative macaques. Real time PCR was performed in 6–12 replicates and at least 2 positive replicates were required to be reported as a positive result. Rh13.1- and Rh157.5-specific primers/probe sets were used in a 15µL reaction with the TaqMan Fast Advanced Master Mix (Applied Biosystem). The reactions were performed in triplicates in 384-well plates for real time quantification using QuantStudio 7 Flex Real-Time PCR Systems (Applied Biosystems) and data were collected using the QuantStudio Real-Time PCR Software v1.3. Copy numbers were calculated relative to 6-fold dilutions of standard plasmids containing the target sequence of each indicated gene. All results were expressed as RhCMV DNA copy number per mL in plasma and per µg input DNA in saliva and urine. Plasma viral load data in the FL-RhCMV inoculated Indian origin RM was compared with historical control CMV-seronegative pregnant female macaques inoculated with RhCMV clinical isolate strains UCD52, UCD59, and 180.92 as previously reported [11].

## PCR amplification and next generation sequencing of RhCMV genomic DNA fragments from urine samples

We designed eight individual primer pairs shown in the **S2 Table** to PCR amplify the RhCMV genomic regions that were altered during the construction of FL-RhCMV (**Fig 1**). Urine samples were collected at 57, 64, or 65 days post infection from RM inoculated with FL-RhCMV/Rh13.1apt. All PCR reactions were performed using the Phusion high-fidelity Hot Start II DNA Polymerase (ThermoFischer Scientific) following the manufacturer's instructions. The amplified DNA fragments were first purified using the GeneJET Gel Extraction Kit (ThermoFisher Scientific) according to the manufacturer's instructions and subsequently used to create

NGS libraries as described above. Next generation sequencing was performed on an iSeq 100 next-generation sequencing platform (Illumina) and all generated sequencing reads were first quality controlled and trimmed and subsequently aligned to the FL-RhCMV genome by reference-guided assembly using the Geneious Prime software. Manual SNP-calling was performed on all single nucleotide polymorphisms (SNPs) with a frequency above 1% of the aligned reads. As a pre-infection control we included PCR fragments amplified from viral DNA isolated from the FL-RhCMV/Rh13.1apt stock used to inoculate the RM in this study. This stock was grown on rhesus fibroblast under tetracycline selection and a full genome NGS analysis of the complete viral genome is shown in **Fig 4C**.

## Nested real-time PCR

To examine the differences in dissemination between FL-RhCMVΔRh13.1/TB6Ag and FL-RhCMVΔRh13.1/TB6AgΔRh157.4–5ΔRh158-161, we infected three RhCMV-naïve RM per vector s.c. with $10^7$ PFU. At 14 days post infection, we took the animals to necropsy and harvested tissues from which the DNA was isolated by the ONPRC Molecular Virology Support Core (MVSC) using the FastPrep (MP Biomedicals) in 1 ml TriReagent (Molecular Research Center Inc.) for tissue samples under 100 mg. Additionally, 100 μl bromochloropropane (MRC Inc.) was added to each homogenized tissue sample to enhance phase separation. 0.5 ml DNA back-extraction buffer (4 M guanidine thiocyanate, 50 mM sodium citrate, and 1 M Tris) was added to the organic phase and interphase materials, which was then mixed by vortexing. The samples were centrifuged at 14,000 x g for 15 minutes, and the aqueous phase was transferred to a clean microfuge tube containing 240 μg glycogen and 0.4 ml isopropanol and centrifuged for 15 minutes at 14,000 x g. The DNA precipitate was washed twice with 70% ethanol and resuspended in 100 to 500 μl double deionized water. Nested real-time PCR was performed with primer and probe sets for the inserted *Mtb* TB6Ag transgene (first round: for-CAGCCGCTACAGATGGAGAG and rev-CGCGCTAGGAGCAAATTCAC; second round: for-CAGCCGCTACAGATGGAGAG, rev-CGCGCTAGGAGCAAATTCAC, and probe-TGGCGGCTTGCAAT-FAM). For each DNA sample, 10 individual replicates (5 μg each) were amplified by first-round PCR synthesis (12 cycles of 95˚C for 30 seconds and 60˚C for 1 minute) using Platinum Taq in 50 μl reactions. Then, 5 μl of each replicate was analyzed by nested quantitative PCR (45 cycles of 95˚C for 15 seconds and 60˚C for 1 minute) using Fast Advanced Master Mix (ABI Life Technologies) in an ABI StepOnePlus Real-Time PCR machine. The results for all 10 replicates were analyzed by Poisson distribution and expressed as copies per cell equivalents [86].

## Histopathology and in situ hybridization (ISH)

RNAscope was performed on formaldehyde fixed, paraffin-embedded tissue sections (5μm) according to our previously published protocol [87] with the following minor modifications: heat-induced epitope retrieval was performed by boiling slides (95–98˚C) in 1x target retrieval (ACD; Cat. No. 322000) for 30 min., followed by incubation at 40˚C with a 1:10 dilution of protease III (ACD; Cat. No. 322337;) in 1x PBS for 20 min. Slides were incubated with the target probe RhCMV (ACD; Cat. No. 435291) for 2 hours at 40˚C and amplification was performed with RNAscope 2.5 HD Brown Detection kits (ACD; Cat. No. 322310) according to manufacturer's instructions, with 0.5X wash buffer (310091, ACD) used between steps, and developed with Alexa-fluor647 conjugated tyramide. The RhCMV probe consisted of 50zz pairs targeting the RhCMV rh38 (13zz pairs), rh39 (10zz pairs all shared with Rh39) and rh55 (37zz pairs) ORFs originally annotated in 68–1 RhCMV [65]. These ORFs correspond to Rh38.1 (UL22A) and Rh55 (UL33) in the annotation of the RhCMV genome presented here. To remove/inactivate the *in situ*

amplification tree/HRP complexes, slides were microwaved at full power for 1 minute, followed immediately for 15 minutes at 20% power in 0.1% citraconic anhydride with 0.05% Tween-20 buffer. Slides were cooled for 15 minutes in the buffer, then rinsed in ddH2O. Slides were subsequently stained for CD34 (Sigma, Cat. No. HPA036723), at a 1:200 dilution in antibody diluent (1x TBS containing 0.25% casein and 0.05% Tween-20) overnight at 4˚C. Detection was performed using an anti-rabbit polymer HRP conjugated system (GBI Labs; Cat. No. D39-110), and developed with Alexa-fluor488 conjugated tyramide at a 1:500 dilution for 10 minutes. To remove the antibody/HRP complexes, a second round of microwaving was performed as described above. Slides were subsequently stained for myeloid lineage cells using a combination of mouse anti-CD68 (Biocare Medical; Cat. No. CM033C; clone KP1) and mouse anti-CD163 (Thermo-Fisher; Cat. No. MA5-1145B; clone 10D6), at a 1:400 dilution (each) in antibody diluent for one hour at room temperature. Detection was performed using an anti-mouse polymer HRP conjugated system (GBI Labs; Cat. No. D37-110), and developed with Alexa-fluor350 conjugated tyramide at a 1:50 dilution for 15 minutes. A third round of microwaving was performed to remove the antibody/HRP complexes as described above. Slides were subsequently stained for mesenchymal lineage cells using a mouse anti-vimentin (Sigma; Cat. No. HPA001762), at a 1:1000 dilution in antibody diluent overnight at 4˚C. Detection was performed using an anti-mouse polymer HRP conjugated system (GBI Labs; Cat. No. D12-110), and developed with Alexa-fluor594 conjugated tyramide at a 1:500 dilution for 10 minutes. To ensure that that HRP inactivation and antibody stripping was complete, matched slides that had gone through the previous staining steps did not receive the subsequent primary antibody, but were developed with all slides from that round. In each case we did not see staining with Alexa-fluor 488, Alexa-fluor 350 or Alexa-fluor 594 tyramide, indicating that the microwave step completely removed/inactivated the *in situ* amplification tree/HRP complexes and removed all previous antibody/HRP complexes. Slides were coverslipped using Prolong Gold antifade mounting media (ThermoFisher; Cat. No. P36930), scanned using a Zeiss AxioScan Z1, and analyzed using Halo software (v2.3.2089.52; Indica Labs). Cell counts were quantified using the FISH Multiplex RNA v2.1 module and RhCMV RNA percent area quantification was performed using the Area Quantification FL v1.2 module. To calculate viral copy number, we used the HALO analysis program to analyze the average size (area) and fluorescent intensity of more than 10 individual virions within the spleen, which was used that to calculate a minimum estimate of RhCMV RNA copy numbers in the infected cells using the FISH Multiplex RNA v2.1 module. Statistical analysis was performed with GraphPad Prism v.7.04. Data are mean +/- S.E.M. as indicated.

## Intracellular cytokine staining (ICS)

Our intracellular cytokine staining (ICS) protocol to examine RM T cell responses has been described previously [14,88]. Briefly, we isolated mononuclear cells from RM tissues collected at necropsy and measured specific CD4[+] and CD8[+] T cell responses by flow cytometric ICS. For this purpose, the isolated cells were incubated with mixtures of consecutive 15mer peptides (overlapping by 11AA) of the *Mtb* TB6Ag in the presence of the costimulatory molecules CD28 and CD49d (BD Biosciences) for 1 hour, followed by addition of brefeldin A (Sigma-Aldrich) for an additional 8 hours. As a background control we used cells co-stimulated without the peptide pool. Following incubation, cells were stored at 4˚C until staining with combinations of fluorochrome-conjugated monoclonal antibodies including: anti-CD3 (SP34-2: Pacific Blue; BD Biosciences), anti-CD4 (L200: BV510; Biolegend), anti-CD8α (SK1: TruRed; eBioscience), anti-TNF-α (MAB11: PE; Life Tech), anti-IFN-γ (B27: APC; Biolegend) and anti-CD69 (FN50: PE/Dazzle 594; Biolegend), anti-Ki67 (B56: FITC; BD Biosciences). Data were collected on an LSR-II flow cytometer (BD Biosciences). FlowJo software (Tree Star) was

used for data analysis. In all analyses, gating on the small lymphocyte population was followed by the separation of the CD3[+] T cell subset and progressive gating on CD4[+] and CD8[+] T cell subsets. Antigen-responding cells in both CD4[+] and CD8[+] T- cell populations were determined by their intracellular expression of CD69 and either or both of the cytokines interferon gamma (IFN-γ) and tumor necrosis factor alpha (TNF-α). Final response frequencies shown have been background subtracted and memory corrected as previously described [17].

### RNA-seq library preparation, sequencing and data analysis

Total RNA was isolated from *in vitro* cell cultures and *in vivo* tissues using the Trizol method. RNA next generational sequencing (NGS) was performed on polyA-fractionated RNA utilizing the TruSeq Stranded mRNA library prep kit (Illumina). The library was validated using Agilent DNA 1000 kit on bioanalyzer according to manufacturer's protocol. RNA libraries were sequenced by the OHSU Massively Parallel Sequencing Shared Resource Core on a Illumina HiSeq-2500 using single-end 100 bp reads. Due to low fraction of viral reads relative to host, the libraries from parotid and submandibular glands were sequenced again to increase read depth. Sequence data were quality trimmed with Trimmomatic [89], and aligned to a custom reference genome comprised of the latest rhesus macaque genome build (MMul10; assembly ID GCA_003339765.3) and the FL-RhCMVΔRh13.1/TB6Ag genome using the STAR aligner [90]. For coverage analyses, GATK DepthOfCoverage [91] was used to produce a table of raw counts per base. Coverage across the RhCMV genome was visualized in R (3.6.1) for the three *in vitro* datasets (8, 24, and 72 hour post infection) using Gviz (1.3) [92]. Base-pair level coverage data was smoothened using the ksmooth function of base R, a kernel-based regression smoothing technique, with the bandwidth parameter set to 500. Feature counting was performed using Subread featureCounts version 1.6.0 [93], using a gene transfer format (GTF) file produced by concatenating Rhesus macaque Ensembl reference genes (build 98) with the ORFs annotated from FL-RhCMVΔRh13.1/TB6Ag. For subread, the following options were used: fracOverlap of 0.1, minOverlap of 20, using both—largestOverlap, and—primary. To account for gene length differences that can bias the transcript counts, we normalized across the genes, for all samples by the gene length (computed as the total exon length from start to end positions). Finally, for an equivalent comparison across the *in vitro* and *in vivo* samples, used for the principal component analysis (PCA) as well as the combined heatmap (**Fig 8A and 8B**), we corrected for cross-sample variation by quantile normalization of the gene expression matrix. PCA was performed using the base R prcomp function on the normalized gene expression matrix as described in the data analysis section. The variance across the components was used to order and select for top components of interest.

## Supporting information

**S1 Fig. Genome organization of NHP CMVs.** The genome of HCMVs and the closely related CCMV comprise two unique coding regions (U_L and U_S) that are separated by an internal repeat region and flanked by terminal repeats. The repeat regions consist of the three repeated sequence units a, b and c that form overlapping inverted repeats in the form *ab-U_L-b'a'c-U_S-c'a*. The HCMV genome can re-arrange to four different isomers varying in the relative orientation of the U_L and U_S regions to one another [109]. Intriguingly, while the U_L and U_S regions can still be identified in old world NHP CMVs, the repeat organization is vastly different. The terminal direct repeats in these species are short while the internal repeats are completely missing resulting in a single isomer that has been fixed during evolution. All Asian NHP CMVs and the African green monkey (Simian) CMV (SCMV) occur in the same isomeric form whereas the U_S region appears in the opposite orientation to the U_L region in the closely related BaCMV and DrCMV. New world (NW) CMVs retained a genome organization with

terminal and internal repeats similar HCMV, but the repeats are organized as non-overlapping inverted repeats flanking the $U_L$ and $U_S$ regions, allowing for isomerization.
(PDF)

**S2 Fig. The RL11 gene family of NHP CMVs.** A) Phylogenetic tree of RL11 family genes from representatives of each NHP CMV species. B) ORF structure of the RL11 family genes in each NHP CMV species. Each gene is color-coded using the same colors as in A) showing the presence/absence of each ORF in a given NHP CMV species.
(PDF)

**S3 Fig. The UL83 (pp65) gene family of NHP CMVs.** A) Phylogenetic tree based on the protein sequences of NHP CMV genes homologous to HCMV UL83 encoding the major tegument protein pp65 from representatives of each NHP CMV species. B) ORF structure of the UL83 family genes in each NHP CMV species. Each gene is color-coded using the same colors as in A) showing the presence/absence of each ORF in a given NHP CMV species.
(PDF)

**S4 Fig. The UL146/UL147 gene family of NHP CMVs.** A) Phylogenetic tree based on the protein sequences of NHP CMV genes homologous to HCMV chemokine-like genes UL146 and UL147 from representatives of each NHP CMV species. B) ORF structure of the UL146/147 family genes in each NHP CMV species. Each gene is color-coded using the same colors as in A) showing the presence/absence of each ORF in a given NHP CMV species.
(PDF)

**S5 Fig. The Rh166 gene family of NHP CMVs.** A) Phylogenetic tree based on the protein sequences of Rh166 family genes from representatives of each NHP CMV species. B) ORF structure of the Rh166 family genes in each NHP CMV species. Each gene is color-coded using the same colors as in A) showing the presence/absence of each ORF in a given NHP CMV species.
(PDF)

**S6 Fig. The US12 gene family of NHP CMVs.** A) Phylogenetic tree based on the protein sequences of NHP CMV genes homologous to the HCMV US12 family encoding seven transmembrane proteins from representatives of each NHP CMV species. B) ORF structure of the US12 family genes in each NHP CMV species. Each gene is color-coded using the same colors as in A) showing the presence/absence of each ORF in a given NHP CMV species.
(PDF)

**S7 Fig. The US28 gene family of NHP CMVs.** A) Phylogenetic tree based on the protein sequences of NHP CMV genes homologous to HCMV US28 encoding G-protein coupled receptors from representatives of each NHP CMV species. B) ORF structure of the US28 family genes in each NHP CMV species. Each gene is color-coded using the same colors as in A) showing the presence/absence of each ORF in a given NHP CMV species.
(PDF)

**S8 Fig. The FL-RhCMV genome remains stable in vivo after inoculation of CMV naïve RM.** Schematic overview depicting the eight DNA fragments (red boxes) that were amplified by PCR from urine samples collected at the indicated timepoints from RhCMV naïve RM infected with FL-RhCMV/Rh13.1apt. PCR amplifications of genome fragments from the viral inoculum were included as a pre-infection controls. All viral ORFs encoded in the targeted genomic region are shown in black boxes. The position and frequency of each putative variation was determined by NGS within each corresponding enriched gene region. The location of

each variation with a frequency of 1% or greater relative to the mean coverage of that particular gene region is graphed under the map of the corresponding locus. All variation are labeled with lower case letters ranked by their position in the genomic fragment. All results are shown in comparison to the clonal FL-RhCMV/Rh13.1apt BAC sequence.
(PDF)

**S9 Fig. Exploratory analysis to assess equivalency across RNA-seq data.** To account for equivalency across the RNA-seq samples, A) background expression was assessed using house-keeping genes ("ACTG1", "RPS18", "MRPL18", "TOMM5", "YTHDF1", "TPT1", "RPS27") [110] which did not identify a specific trend across samples. Next, B) host (RM) library size was assessed across samples which showed higher overall transcript levels that were tissue specific to the salivary glands. However, this difference did not correlate with the viral gene expression levels. Finally, C) we found that all three in vitro samples had a higher number of total viral reads than tissue biopsies.
(PDF)

**S10 Fig. Sequence coverage of FL-RhCMVΔRh13.1/TB6Ag in cultured fibroblasts.** Sequence coverage of FL-RhCMVΔRh13.1/TB6Ag in cultured fibroblasts, infected *in vitro* and sampled at three timepoints post-infection (8, 24, 72 hours). Coverage per base is plotted with a log10 scale. Colors: blue = 8hpi, magenta = 24hpi, green = 72hpi.
(PDF)

**S1 Table. List of all RhCMV ORFs affected by changes made to the previously published genome annotation using comparative genomics.**
(DOCX)

**S2 Table. List of primers used to amplify eight gene fragments from viral DNA samples isolated from the urine of FL-RhCMV/Rh13.1apt-infected RM.**
(DOCX)

**S3 Table. Absolute number of RNAseq reads aligning to annotated RhCMV ORFs for all *in vitro* and *in vivo* samples.**
(XLSX)

## Acknowledgments

We would like to thank Dr. Thomas Shenk (Princeton University) for providing the 68–1.2 RhCMV BAC and the rhesus retinal pigment epithelium (RPE) cells. We would also like to thank the faculty and staff of the Departments of Veterinary Medicine and Collaborative Research at TNPRC for the excellent animal care and collection of research specimens. We are grateful to the OHSU molecular virology support core for generating RhCMV stocks and for assisting in the processing of tissue samples, to Dr. Robert Searles and the OHSU Massively Parallel Sequencing Shared Resource (MPSSR) for generating the NGS libraries and to Yibing Jia and the Molecular and Cellular Biology Core (MCB Core) at the ONPRC for analyzing the samples on their MiSeq next generation sequencer. Lastly, we would like to acknowledge the invaluable technical and scientific support provided by Katinka Vigh-Conrad, Larry Wilhelm, Shoko Hagen, Andrew Sylwester, Kyle Taylor, Junwei Gao, Jennie Womack and Nessy John that made this study possible.

## Author Contributions

**Conceptualization:** Husam Taher, Eisa Mahyari, Benjamin N. Bimber, Louis J. Picker, Daniel N. Streblow, Klaus Früh, Daniel Malouli.

**Data curation:** Husam Taher, Eisa Mahyari, Scott G. Hansen, Amitinder Kaur, Benjamin N. Bimber, Louis J. Picker, Daniel N. Streblow, Daniel Malouli.

**Formal analysis:** Husam Taher, Eisa Mahyari, Michael Nekorchuk, Jason Shao, Paul T. Edlefsen, Scott G. Hansen, Benjamin N. Bimber, Daniel Malouli.

**Funding acquisition:** Louis J. Picker, Klaus Früh.

**Investigation:** Husam Taher, Eisa Mahyari, Craig Kreklywich, Luke S. Uebelhoer, Matthew R. McArdle, Matilda J. Moström, Amruta Bhusari, Michael Nekorchuk, Xiaofei E, Travis Whitmer, Elizabeth A. Scheef, Lesli M. Sprehe, Dawn L. Roberts, Colette M. Hughes, Kerianne A. Jackson, Andrea N. Selseth, Abigail B. Ventura, Hillary C. Cleveland-Rubeor, Yujuan Yue, Kimberli A. Schmidt, Timothy F. Kowalik, Richard J. Stanton, Scott G. Hansen, Benjamin N. Bimber, Daniel Malouli.

**Methodology:** Husam Taher, Eisa Mahyari, Benjamin N. Bimber, Louis J. Picker, Klaus Früh, Daniel Malouli.

**Project administration:** Richard J. Stanton, Scott G. Hansen, Amitinder Kaur, Daniel N. Streblow, Daniel Malouli.

**Resources:** Jeremy Smedley, Timothy F. Kowalik, Michael K. Axthelm, Jacob D. Estes, Scott G. Hansen, Amitinder Kaur, Peter A. Barry, Benjamin N. Bimber, Louis J. Picker.

**Software:** Eisa Mahyari, Benjamin N. Bimber.

**Supervision:** Scott G. Hansen, Daniel N. Streblow, Daniel Malouli.

**Validation:** Husam Taher, Eisa Mahyari, Benjamin N. Bimber, Daniel Malouli.

**Visualization:** Husam Taher, Eisa Mahyari, Matilda J. Moström, Michael Nekorchuk, Scott G. Hansen, Benjamin N. Bimber, Daniel Malouli.

**Writing – original draft:** Klaus Früh, Daniel Malouli.

**Writing – review & editing:** Husam Taher, Eisa Mahyari, Matilda J. Moström, Xiaofei E, Timothy F. Kowalik, Richard J. Stanton, Jacob D. Estes, Scott G. Hansen, Amitinder Kaur, Peter A. Barry, Benjamin N. Bimber, Louis J. Picker, Daniel N. Streblow, Klaus Früh, Daniel Malouli.

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
