## [Decision Letter · Decision Letter 0]

29 Jun 2020

Dear Dr. Malouli,

Thank you very much for submitting your manuscript "In vitro and in vivo characterization of a recombinant rhesus cytomegalovirus containing a complete genome" for consideration at PLOS Pathogens. As with all papers reviewed by the journal, your manuscript was reviewed by members of the editorial board and by several independent reviewers. The reviewers appreciated the attention to an important topic. Based on the reviews, we are likely to accept this manuscript for publication, providing that you modify the manuscript according to the review recommendations.

Sincerely,

Robert F. Kalejta

Associate Editor

PLOS Pathogens

Shou-Jiang Gao

Section Editor

PLOS Pathogens

Kasturi Haldar

Editor-in-Chief

PLOS Pathogens

orcid.org/0000-0001-5065-158X

Michael Malim

Editor-in-Chief

PLOS Pathogens

orcid.org/0000-0002-7699-2064

Reviewer Comments (if any, and for reference):

Reviewer's Responses to Questions

**Part I - Summary**

Reviewer #1: In their manuscript titled “In vitro and in vivo characterization of a recombinant rhesus cytomegalovirus containing a complete genome,” Taher and colleagues describe the repair of all the gene defects in RhCMV 68-1. Then, they characterize the full length RhCMV (FL-RhCMV) both in vitro and in vivo. In light of the potential value of CMV-based vaccines, it is important to understand how a fully intact CMV virus would replicate in rhesus macaques. Much of the vaccine work has been performed using the defective CMV viruses. While those pioneering studies greatly advance the field of infectious disease vaccine research, a more thorough understanding of how CMV can be manipulated to be both safe and immunogenic is necessary for vaccine development. By generating this full length rhesus CMV, engineering some deletions, and then assessing the impact of those, these authors have generated an essential reagent in the toolbox needed to develop a clinically relevant CMV-based vaccine.

These authors clearly demonstrate a large amount of work in this manuscript, and they should be commended for it. There are, however, several points in the text that need clarification. Right now, it is difficult to read, and I think that some of the key findings get buried behind other pieces of data that seem tangential.

Overall, this is a study that will provide a nice addition to the field. It is important to describe the characterization of a FL-RhCMV and identify the impact of specific mutations on function of this virus. This paper, however, feels like there is extra data included that is not relevant to a discussion of the construction of the virus. This paper would greatly benefit from removing extraneous data and telling a clear and linear narrative about how they made the FL-RhCMV, how it replicates, and then how specific mutations affect its function.

Reviewer #2: In this paper by Malouli and colleagues, a recombinant BAC with repairs in a cluster of rhesus CMV genes is generated. The BAC construct restores virulence to the virus. This is a big advance in the study of RhCMV and will enable the study of pathogenesis in this important animal model. The paper will be of interest to basic scientists, molecular virologists, and translational scientists in internal medicine and pediatric infectious diseases.

The paper is well written and has appropriate controls. Conclusions are supported by the data. Inoculation of monkeys with the reconstituted virus resulted in significant replication in the blood similar to primary isolates of RhCMV and end-organ disease. Viral dissemination and viremia was greatly reduced upon deletion of genes also lacking in 68-1. Transcriptome analysis of infected tissues further revealed that chemokine genes deleted in 68-1 were highly expressed, consistent with an immune modulation effect.

It is presumed from this work that these genes PROMOTE inflammation which facilitates dissemination, and are not for IMMUNE EVASION per se. Thus, RhCMV, like human CMV, "loves an inflammatory party".

In addition to the usefulness of the new BAC and the insights it provides into pathogenesis, the paper is an elegant treatise on molecular phylogeny of CMVs and is very useful in-and-of-itself for this reason.

A few suggestions are offered toward the goal of improving the paper:

1. It is a bit of a misnomer to speak of extensive "viremia in many tissues". Viremia is in the blood; virus in the tissue is in the tissue, not in the blood.

2. Lines 571-573, "...supernatants from cells that were spin-inoculated with the original urine sample were collected at full CPE and used to infect three T-175 flasks of primary rhesus fibroblasts". Recognizing the difficulty in amplifying CMV, even a single passage on fibroblasts could result in genome modifications and mutations. Did the authors try NGS directly from the urine sample? Can the authors provide reassurance that the passage did not modify the parental sequence?

3. Lines 777-782, it seems surprising that given the exceptional sensitivity of real-time PCR that a nested step would be require. Did the authors fail to see real-time PCR signal with single-primer-pair real-time PCR? Did the authors attempt digital PCR? Some comment on why nested PCR was necessary would be useful.

4. Lines 225-227, "...Loss of RL13 could be prevented in HCMV by conditional expression of RL13 mRNA under the control of a tet operator (tetO) and growth in tet-repressor (tetR)-expressing fibroblasts." True, but the authors should also point out that changes in RL13 can be obviated by inclusion of antibody in the cell culture media (https://doi.org/10.3390/v11030221). The authors should cite this paper and included a comment on this.

5. The differences in this novel, new BAC and the 681- construct seem clear and well delineated in this paper. It is unclear what the authors believe the role of these UL128-131 homologs in the RhCMV genome really is. The investigative group has published that these ORFS are involved in the generation of novel, new "promiscuous" T cell epitopes; that they form a pentamer that is important in epithelial cell entry; and that there is a chemokine function attributable to some regions of this cluster of gene. It would be helpful if the authors could provide a summative, integrative statement in the discussion as to what function they believe is most important, and the pros/cons of considering this as a truly homologous cluster to HCMV.

6. The authors comment on animal models (line 485) for congenital CMV infection. It is agreed that this is a very important off-shoot of this work. Toward providing a more comprehensive and current list of references, the authors should add this reference to their paper and reference it here: DOI: 10.1093/infdis/jiz484

7. Also along these same lines, it is surprising that the paper does not comment on this paper: doi: 10.1172/jci.insight.94002. In this paper, Nelson et al. The paper is cited but not commented on. In this paper, macaques that were inoculated with fibroblast-tropic 180.92 and epithelial cell–tropic UCD52/UCD59 had disease. Can Malouli and colleagues comment on the pathogenicity of UCD52/UCD59 compared to their fully wild-type BAC virus? If UCD52/UCD59 can cause disease, including congenital infection, what advantages does this full length BAC virus offer?

Reviewer #3: Taher, Mahyari, Malouli and colleagues report about a novel BAC-cloned RhCMV which they have constructed on the basis of the well-established 68-1 strain by restoring a number of deviations that this strain has as compared with primary RhCMV isolates. In detail, the authors repaired Rh13.1(RL13 in HCMV) Rh61/Rh60 (UL36), Rh152/Rh151 (UL119/UL118), Rh157.5 and Rh157.4 (UL128 and UL130), Rh164 (UL141), Rh167, Rh197 (US14D), and this restored RhCMV-BAC they named "full length Rhesus cytomegalovirus" FL-RhCMV. With this work they not only complemented the BAC cloned 68-1 strain on a genetic level, but apparently they also restored biological functions in a way that the new clone resembles primary isolates with regard to in vivo replication in rhesus macaques. This is an outstanding contribution to the field, given the virus is made available to the scientific community without restrictions.

**Part II – Major Issues: Key Experiments Required for Acceptance**

Reviewer #1: 1. Figure 1 shows all the different repairs that were made in 68-1 to generate the FL-RhCMV. The description of the repair steps are in the legend, but they are not explained in the text. The rationale for making each repair is buried. If each repair is important, then a rationale explaining each repair mutant and why only certain mutants were tested in culture would be useful.

2. Several mutations are made in the FL-RhCMV backbone to recapitulate some of the specific deletions in 68-1. While this approach makes a lot of sense, this is not laid out very clearly. A summary of all the different mutations in FL-RhCMV and their rationale would be beneficial. This includes the deletion of Rh13.1 and the various sub mutations or insertions into that backbone. Each of these mutants seems to be just a little bit different and their purposes are different. It seems like a variety of mutations were made (for good reasons) and then just added together in this paper. A linear discussion of why these mutations were made and what hypothesis each one tested is needed for the reader to better understand the manuscript.

3. The transcript analysis seems like a herculean amount of work, but I cannot tell how that impacts our understanding of the construction of the full length RhCMV. This needs to be explained more clearly. Was this section intended to measure expression of CMV genes or assess how the CMV genes would modulate host immunity?

4. The phylogenetic relationship between FL-RhCMV and other NHP CMV strains also seems inessential for this study.

Reviewer #2: This reviewer has no criticism of any experiments nor any new suggestions for new experiments. This is elegant and well-controlled work.

Reviewer #3: The data presented in Fig. 5A are very surprising, given the strong inhibitory effect of RH13.1 shown in figure 4B that also suggested an effect on cell-free virus spread (note the lack of dispersed infected cells).

Hence we doubt whether FL-RhCMV in the experiments of 5A had an intact ORF Rh13.1 during the complete course of the growth curve. In HCMV, it is known that RL13 already is disrupted during reconstitution of the virus by transfection of the BAC. It is conceivable that RH13.1 became disrupted during initial replication and was selected during the second and/or third replication cycle of this multistep growth curve.

The authors must at minimum sequence the virus harvested on day 12 to check whether RH13.1 was still intact.

This is particularly important in the context of the authors' remark that "our results indicate that FL-RhCMV is remarkably similar to low passage clinical isolates of HCMV with respect to growth in tissue culture, tissue tropism and genetic stability in vitro." (lines 285-287). HCMV isolates grow strictly cell-associated in relatively small foci, which resembles FL-RhCMV in TRF (Fig 4B) but is in sharp contrast to the strong cell-free growth in Fig 5A.

Similarly, sequence data should be obtained regarding the samples in in-vivo experiments to check whether the restored gene regions are still intact. In my opinion, this is at least mandatory for those gene regions which are functionally addressed in this paper, i.e. RH13.1; the PRC genes, and so forth.

**Part III – Minor Issues: Editorial and Data Presentation Modifications**

Reviewer #1: There are typos and acronyms not spelled out.

Reviewer #2: See comments above. Generally, exceptionally well written and clear manuscript.

Reviewer #3: 216: "is likely a representative of the genomes contained in the original 68-1 isolate" should be further toned down, as the original sequence is not available for comparison => e.g. "expected to resemble" "expected to approximate"

249: "mutations in the Rh13.1 homologs found in many old world NHP CMV genomes (Fig. 3) are due to"; this reference to Fig. 3 does not make much sense as no sequence data are shown there.

259: Typo: "we cannot observed a phenotype" => observe

264: "In contrast, a FL-RhCMV vector carrying an SIVgag insert replacing Rh13.1 (FL-RhCMVΔRh13.1gag) displayed an increased ability to enter RPE cells compared to 68-1." We miss data and/or information regarding this virus.

In Figure 5A, I do not find it accurate to show these sem bars. Obviously, they are the result of two technical replicates from two biological replicates, which must not mixed to yield "n=4" in my opinion. So, I suggest to leave it with means of two biological replicates. If the authors really want to improve these data, they may add one further biological replicate. As the data appear at the moment, there might be a significant difference (about 10fold) between 68-1 and 68-1.2, which can, however, only be tested after an additional experiment. NOTE: this remark only applies if sequence data show that RH13.1 and the PRC gene region are intact in the repaired versions, see "major issues".

In Figure 6, panel C does not add information but is rather confusing, due to the fact that the last two time points of "A" are no triplicates. Hence the "indentation" of the curve at around 18 days is meaningless. I suggest to remove this panel and rather include all six curves in one graph, with the curve sets of "A" and "B" in different colors.

PLOS authors have the option to publish the peer review history of their article (what does this mean?). If published, this will include your full peer review and any attached files.

Reviewer #1: **Yes: **Shelby L OConnor

Reviewer #2: No

Reviewer #3: No
---

## [Editor Report · Decision Letter 1]

30 Sep 2020

Dear Dr. Malouli,

We are pleased to inform you that your manuscript 'In vitro and in vivo characterization of a recombinant rhesus cytomegalovirus containing a complete genome' has been provisionally accepted for publication in PLOS Pathogens.

Best regards,

Robert F. Kalejta

Associate Editor

PLOS Pathogens

Shou-Jiang Gao

Section Editor

PLOS Pathogens

Kasturi Haldar

Editor-in-Chief

PLOS Pathogens

orcid.org/0000-0001-5065-158X

Michael Malim

Editor-in-Chief

PLOS Pathogens

orcid.org/0000-0002-7699-2064
---

## [Editor Report · Acceptance letter]

3 Nov 2020

Dear Dr. Malouli,

We are delighted to inform you that your manuscript, "In vitro and in vivo characterization of a recombinant rhesus cytomegalovirus containing a complete genome," has been formally accepted for publication in PLOS Pathogens.

Best regards,

Kasturi Haldar

Editor-in-Chief

PLOS Pathogens

orcid.org/0000-0001-5065-158X

Michael Malim

Editor-in-Chief

PLOS Pathogens

orcid.org/0000-0002-7699-2064